# Features are fate: a theory of transfer learning in high-dimensional regression

**Javan Tahir**[1]   **Surya Ganguli**[1]   **Grant Rotskoff**[2]

## Abstract

With the emergence of large-scale pre-trained neural networks, methods to adapt such "foundation" models to data-limited downstream tasks have become a necessity. Fine-tuning, preference optimization, and transfer learning have all been successfully employed for these purposes when the target task closely resembles the source task, but a precise theoretical understanding of "task similarity" is still lacking. We adopt a *feature-centric* viewpoint on transfer learning and establish a number of theoretical results that demonstrate that when the target task is well represented by the feature space of the pre-trained model, transfer learning outperforms training from scratch. We study deep linear networks as a minimal model of transfer learning in which we can analytically characterize the transferability phase diagram as a function of the target dataset size and the feature space overlap. For this model, we establish rigorously that when the feature space overlap between the source and target tasks is sufficiently strong, both linear transfer and fine-tuning improve performance, especially in the low data limit. These results build on an emerging understanding of feature learning dynamics in deep linear networks, and we demonstrate numerically that the rigorous results we derive for the linear case also apply to nonlinear networks.

## 1. Introduction

State-of-the-art neural network models have billions to trillions of parameters and are trained on datasets of a similar scale. The benefits of dataset scale are manifest in the astounding generalization capability of these foundation models (Bahri et al., 2024). For many applications, however, datasets of the scale used for natural language processing or computer vision are difficult, if not impossible, to generate. To alleviate the problem of inadequate dataset scale, the representations of a foundation model seem to provide a useful inductive bias for adaptation to a target task. While they are now ubiquitous, *transfer learning* methods lack a solid theoretical foundation or algorithmic design principles. As such, it remains difficult to predict when—and with which approach—transfer learning will outperform training on the target task alone. By studying an analytically solvable model, we find that the feature space learned during pretraining is the relevant object for predicting transfer performance. Of course, adopting a feature-centric viewpoint creates model-specific challenges because unambiguously identifying learned features remains an outstanding and difficult characterization problem for deep neural networks. For this reason, in this work we focus on deep linear networks trained with gradient flow, as feature learning dynamics are well-understood in this setting[1]. We develop an intuitive understanding of linear transfer and full fine-tuning in this model. In contrast to other recent work, we quantify transfer performance relative to training on the target task alone and precisely identify when transfer learning leads to improved performance, effectively building a phase diagram for transfer efficiency. Finally, in numerical experiments, we show that this picture holds qualitatively for nonlinear networks as well.

### Related Work

**Theoretical aspects of transfer learning**   A number of recent works have studied theoretical aspects of transfer learning, focusing on the risk associated with various transfer algorithms. Wu et al. (2020) use information theory to derive bounds on the risk of transfer learning using a mixture of source and target data. Shilton et al. (2017) analyze transfer in the context of Gaussian process regression. Tripuraneni et al. (2020) work in a fairly general setting, and derive bounds on the generalization error of transferred models through a complexity argument, highlighting the importance of feature diversity among tasks. Aminian et al. (2024) study transfer learning in highly overparameterized

[1]Department of Applied Physics, Stanford University, Stanford CA, USA [2]Department of Chemistry, Stanford University, Stanford CA, USA. Correspondence to: Javan Tahir <javan@stanford.edu>.

*Proceedings of the $42^{nd}$ International Conference on Machine Learning*, Vancouver, Canada. PMLR 267, 2025. Copyright 2025 by the author(s).

---

[1]All code to reproduce the results in this paper can be found at https://github.com/javantahir/features_are_fate

models, including one hidden layer neural networks, and derive bounds on the excess risk. Bu et al. (2021) study the excess risk of transferred models optimized with the Gibbs algorithm and highlight a bias-variance interpretation of the generalization performance. Liu et al. (2019); Neyshabur et al. (2020) study transfer learning from the perspective of the loss landscape and find that transferred models often find flatter minima than those trained from scratch. Consistent with our feature-centric viewpoint, Kumar et al. (2022) show that fine-tuning can distort the pretrained features, leading to poor out of distribution behavior. Integral probability metrics on the dataset and target have been used to predict transfer learning performance. For example, Wu et al. (2020); Nguyen et al. (2021); Alvarez-Melis & Fusi (2020) suggest that these metrics correlate with generalization error on the target task. However, we prove in Appendix A, that integral probability metrics such as the Wasserstein-1 Distance, Dudley Metric and $\phi-$divergences such as the KL-divergence need not correlate with transfer efficiency, emphasizing that transferrability is a property of the learned feature space and not the source and target datasets alone.

**Transfer learning in solvable models**  Similar to our approach, several theory works have worked with analytically tractable models to more precisely characterize transfer performance. Lampinen & Ganguli (2018); Atanasov et al. (2021); Shachaf et al. (2021) also study transfer learning in deep linear networks, but focus on the generalization error alone, not the transferability relative to a scratch trained baseline, which obfuscates the conditions for transfer learning to be beneficial. Gerace et al. (2022) studies transfer learning with small nonlinear networks with data generated from a "hidden manifold" (Goldt et al., 2020) and find it to be effective when tasks are very similar, and data is scarce, but do not theoretically describe regions of negative transfer. Saglietti & Zdeborova (2022) studies knowledge distillation in a solvable model, which can be viewed as a special case of transfer learning. Ingrosso et al. (2024) study transfer learning in a model similar to ours using the replica method and similarly conclude that a feature-based metric for task similarity is predictive of transfer performance.

**Feature learning**  The feature space of a neural network evolves in the feature learning regime, which should be contrasted with the static neural tangent kernel limit, in which networks regress functions onto a set of basis functions which are fixed at initialization (Jacot et al., 2018). Recently, there has been an explosion of interest in understanding dynamics in these two regimes of neural network optimization. Woodworth et al. (2020); Atanasov et al. (2021); Yang & Hu (2021); Kunin et al. (2024); Yun et al. (2021); Chizat (2020) focus on the role of initialization, learning rate, and implicit bias in feature learning. Petrini et al. (2022) highlights the potential for overfitting when training in the feature learning

regime. Allen-Zhu & Li (2020a;b) give a fairly complete treatment of feature learning for synthetic datasets whose features are readily identifiable. Chen et al. (2023) study a synthetic dataset closely related to that in Allen-Zhu & Li (2020a) and describe how generalizable features are important to out of distribution generalization, a conclusion we also draw in the context of transfer learning.

**Our contributions**

- Since transfer performance depends on the learned representation of source task, we study deep linear networks trained on a linear target function, a setting in which training dynamics, implicit bias, and generalization error can be described rigorously.

- Within this model, we analytically compute a phase diagram that illustrates how transfer learning performs relative to training from scratch on a given task.

- We prove that simple diagnostics, such as distributional measures of source-target distance are insufficient for predicting the success of transfer learning and advance the idea that task similarity should be measured in the space of features instead.

- We also compute the transfer phase diagram for nonlinear neural networks and show that the same picture applies to the reproducing kernel Hilbert space (RKHS) associated with the nonlinear features of the pre-trained network.

## 2. General theoretical setting

We begin by introducing the general theoretical framework under which we study transfer learning. We consider *source* and *target* regression tasks defined by probability distributions $p_s(\boldsymbol{x}, y)$ and $p_t(\boldsymbol{x}, y)$ over inputs $\boldsymbol{x} \in \mathbb{R}^d$ and labels $y \in \mathbb{R}$. We focus on *concept drift*, in which $p_s(\boldsymbol{x}, y) = p(\boldsymbol{x})p_s(y \mid \boldsymbol{x})$ and $p_t(\boldsymbol{x}, y) = p(\boldsymbol{x})p_t(y \mid \boldsymbol{x})$ for the same input distribution $p(\boldsymbol{x})$. The labels are generated from noisy samples of source and target functions $y_s = f_s^*(\boldsymbol{x}) + \epsilon_s$ and $y_t = f_t^*(\boldsymbol{x}) + \epsilon_t$ where $f_s^*(\boldsymbol{x}), f_t^*(\boldsymbol{x}) \in L_2(p(\boldsymbol{x}))$ and $\epsilon_s, \epsilon_t \sim \mathcal{N}(0, \sigma^2)$. During *pretraining*, we train a model with parameters $\boldsymbol{\Theta} = (\boldsymbol{c}, \boldsymbol{\theta})$ of the form

$$f(\boldsymbol{x}; \boldsymbol{\Theta}) = \sum_{i=1}^{m} c_i \phi_i(\boldsymbol{x}; \boldsymbol{\theta}) \qquad (1)$$

on the source task using a mean squared loss. Note that the features $\phi(\boldsymbol{x}, \boldsymbol{\theta})$ are left general and could for example represent final hidden activations of a deep neural network. After pretraining, the model is transferred by training a subset of the parameters $\boldsymbol{\Theta}' \subset \boldsymbol{\Theta}$ on the target task, while leaving $\boldsymbol{\Theta} - \boldsymbol{\Theta}'$ at their pretrained values. To model the

modern setting for transfer learning, in which the number of data points in the source task far exceeds those in the target task, we train the source task on the population distribution and the target task on a finite dataset $\mathcal{D}$ of $n$ independent samples.

$$\mathcal{L}_s(\boldsymbol{\Theta}) = \frac{1}{2}\mathbb{E}_{p_s(\boldsymbol{x},y)}[(f(\boldsymbol{x},\boldsymbol{\Theta}) - y)^2] \qquad (2)$$

$$\mathcal{L}_t(\boldsymbol{\Theta}') = \frac{1}{2}\hat{\mathbb{E}}_{p_t(\boldsymbol{x},y)}[(f(\boldsymbol{x},\boldsymbol{\Theta}) - y)^2] \qquad (3)$$

where $\hat{\mathbb{E}}_p(h(\boldsymbol{x},y)) = \frac{1}{n}\sum_{i=1}^{n} h(\boldsymbol{x}_i, y_i)$ is the expectation over the empirical distribution of $\mathcal{D}$. We focus on two widely employed transfer methods, *linear transfer* and *fine-tuning*. In linear transfer, the pretrained features $\phi(\boldsymbol{x},\boldsymbol{\theta})$ are frozen and only the output weights $\boldsymbol{c}$ are trained on the target task. In fine-tuning, the entire set of parameters $\boldsymbol{\Theta}$ are trained from the pretrained initialization on the target task. In Appendix C, we explore the case when the source dataset is also finite, and show that for fine-tuning, the qualitative picture discussed below is unchanged. To optimize the loss functions (2) and (3), we use gradient flow,

$$\frac{d\boldsymbol{\Theta}_i}{dt} = -\nabla_{\boldsymbol{\Theta}_i}\mathcal{L}(\boldsymbol{\Theta}), \qquad (4)$$

where we have set the learning rate equal to unity for the purpose of analysis. To assess the performance of transfer learning we compare the generalization of the transferred model to a *scratch-trained* model with the same architecture (1) trained on the target same data from a random initialization. We introduce the *transferability* to quantify this relationship:

$$\mathcal{T} = \mathbb{E}_{\mathcal{D}}(\mathcal{R}_{sc} - \mathcal{R}_{tx}) \qquad (5)$$

where $\mathbb{E}_{\mathcal{D}}$ is the expectation over iid draws of the training set and $\mathcal{R}_{tx}$ and $\mathcal{R}_{sc}$ are the generalization errors of the transferred model and scratch trained model, respectively, where the generalization error (or population risk) is given by,

$$\mathcal{R} = \mathbb{E}_{p(\boldsymbol{x})}[((f(\boldsymbol{x},\boldsymbol{\Theta}) - f^*(\boldsymbol{x}))^2]. \qquad (6)$$

We consider transfer learning successful when $\mathcal{T} > 0$, i.e., when the expected generalization of transfer learning outperforms training from scratch on the target task. We refer to the situation $\mathcal{T} < 0$ as *negative transfer*, since pretraining leads to degradation of the generalization error.

## 3. Deep linear networks: an exactly solvable model

Because insight into feature learning is required to understand the dynamics of transfer learning, we first consider an analytically solvable model of transfer learning using deep linear networks. This model gives us direct access to the evolution of the time-dependent feature space and its

distortion under various transfer learning schemes. Let $f$ denote a deep linear network with $L$ layers

$$f(\boldsymbol{x}) = \boldsymbol{x}^T \boldsymbol{W}_1 \boldsymbol{W}_2 \ldots \boldsymbol{W}_L \qquad (7)$$

where $\boldsymbol{W}_l \in \mathbb{R}^{d_{l-1} \times d_l}$ for $l \in [1, 2, \ldots L - 1]$ and $\boldsymbol{W}_L \in \mathbb{R}^{d_{L-1} \times 1}$. For notational convenience we have renamed $\boldsymbol{c}$ in (1) as $\boldsymbol{W}_L$ and for simplicity we set $d_0 = d_1 = \cdots = d_{L-1} = d$, the dimension of the data. The parameter matrices are initialized as $\boldsymbol{W}_l(0) = \alpha \bar{\boldsymbol{W}}_l$ where $\alpha \in \mathbb{R}$. The matrices $\boldsymbol{W}_l(0)$ additionally satisfy (19), which is a technical assumption that generalizes common initialization schemes such as He initialization (Yun et al., 2021; He et al., 2015). Since transfer learning relies on learning features in the source task, we initialize the network in the feature learning regime $\alpha \to 0$. In the following, we assume:

**Assumption 3.1.** Assume that the input data $\boldsymbol{x} \in \mathbb{R}^d$ is normally distributed and that each dataset $\mathcal{D}$ consists of $n$ pairs $\{(\boldsymbol{x}_i, y_i)\}_{i=1}^{n}$ sampled iid from $p_t$ with Gaussian label noise of variance $\sigma^2$.

**Assumption 3.2.** We assume that the source and target functions are each linear functions in $L_2(\mathbb{R}^d, p)$; equivalently, $f_s^*(\boldsymbol{x}) = \boldsymbol{\beta}_s^T \boldsymbol{x}$, $f_t^*(\boldsymbol{x}) = \boldsymbol{\beta}_t^T \boldsymbol{x}$ with $\|\boldsymbol{\beta}_s\|_2^2 = \|\boldsymbol{\beta}_t\|_2^2 = 1$.

To control the level of source-target task similarity, we fix the angle $\theta$ between the ground truth source and target functions so that $\boldsymbol{\beta}_s^T \boldsymbol{\beta}_t = \cos\theta$. The source and target loss functions are given by (2) and (3). When training over the empirical loss, it is convenient to work in vector notation $\mathcal{L}_t(\{\boldsymbol{W}_{l \leq L}\}) = \frac{1}{2n}\|\boldsymbol{y}_t - \boldsymbol{X}\boldsymbol{W}_1\boldsymbol{W}_2\ldots\boldsymbol{W}_L\|_2^2$ where $\boldsymbol{X} \in \mathbb{R}^{n \times d}$ and $\boldsymbol{y} \in \mathbb{R}^n$. We study this model in the high dimensional limit in which $\gamma = n/d$ remains constant as $n, d \to \infty$.

Linear networks have the advantage of analytic tractability, but we note that the representation capacity of these models is limited to affine transformations. Furthermore, the expressiveness of the model is independent of the number of layers. As a result, this model may fail to capture aspects of transfer learning that depend strongly on depth separation (Telgarsky, 2016; Daniely, 2017) or other nonlinear phenomena. However, overparameterized linear models, recapitulate many phenomena observed in deep learning, including double descent (Nakkiran et al., 2021; Belkin et al., 2019), scaling laws (Bahri et al., 2024), feature learning (Vyas et al., 2024; Atanasov et al., 2021) and, as we show, the impact of feature learning on transfer efficiency.

### 3.1. Pretrained models represent source features

To describe transfer efficiency in this setup, we need to understand the function that the model implements after training on the source task. We can describe the network in function space by tracking the evolution of $\boldsymbol{\beta}(t) = \boldsymbol{W}_1 \boldsymbol{W}_2 \ldots \boldsymbol{W}_L$ under gradient flow, such that

the network function at any point in the optimization is $f(\boldsymbol{x}; t) = \boldsymbol{\beta}(t)^T \boldsymbol{x}$. The following Lemma establishes that pretraining perfectly learns the source task in the large source data limit.

**Lemma 3.3.** *Under gradient flow (4) on the population risk objective (2) with initialization satisfying (19),* $\lim_{t \to \infty} \boldsymbol{\beta}(t) = \boldsymbol{\beta}_{\mathrm{s}}$

We prove Lemma 3.3 in Appendix D.2. While this result establishes recovery of the ground truth on the source task, it does not describe the feature space of the pretrained model, which is relevant for transferability. To this end, following Yun et al. (2021), we show that in the feature learning regime $\alpha \to 0$, the hidden features of the model sparsify to those present in the source task.

**Theorem 3.4** (Yun et al). *Let the columns of* $\boldsymbol{\Phi} = \boldsymbol{W}_1 \boldsymbol{W}_2 \cdots \boldsymbol{W}_{L-1}$ *denote the hidden features of the model. After pretraining*

$$\lim_{\alpha \to 0} \lim_{t \to \infty} \boldsymbol{\Phi} = \boldsymbol{\beta}_{\mathrm{s}} \boldsymbol{v}_{L-1}^T$$

*for some vector* $\boldsymbol{v}_{L-1}$.

We prove Theorem 3.4 in Appendix D.3. Theorem 3.4 demonstrates that after pretraining in the feature learning regime, the $d$-dimensional feature space of the model parsimoniously represents the ground truth function in a single, rank-one component. We refer to this phenomenon as feature sparsification, which is a hallmark of the feature learning regime, and has important consequences for transferability, particularly in the linear setting Section 3.3.

## 3.2. Scratch trained models represent minimum norm solutions

For the empirical training objective (3), there are multiple zero training error solutions when the model is overparameterized $\gamma < 1$. As noted in Yun et al. (2021) and Atanasov et al. (2021), there is an implicit bias of gradient flow to the minimum norm solution when $\alpha \to 0$

**Theorem 3.5** ( (Yun et al., 2021)). *Under gradient flow on the empirical risk minimization objective (3) with initialization satisfying (19),* $\lim_{\alpha \to 0} \lim_{t \to \infty} \boldsymbol{\beta}(t) = \hat{\boldsymbol{\beta}}$, *where* $\hat{\boldsymbol{\beta}}$ *is the minimum norm solution to the linear least squares problem*

$$\hat{\boldsymbol{\beta}} = \arg \min_{\boldsymbol{\beta}} \frac{1}{2n} \|\boldsymbol{y}_{\mathrm{t}} - \boldsymbol{X} \boldsymbol{\beta}\|_2^2 = \boldsymbol{X}^+ \boldsymbol{y}_{\mathrm{t}}$$

We prove Theorem 3.5 in Appendix D.4. Knowing the final predictor of the empirical training also allows us to compute the generalization error of the scratch trained model

**Theorem 3.6.** *Under gradient flow on the empirical objective (3), in the high dimensional limit the expectation of the*

final generalization error over training data is

$$\mathbb{E}_{\mathcal{D}} \mathcal{R} = \begin{cases} \frac{(1-\gamma)^2 + \gamma \sigma^2}{1 - \gamma} & \gamma < 1 \\ \frac{\sigma^2}{\gamma - 1} & \gamma > 1 \end{cases} \qquad (8)$$

Theorem (3.6) is a known result for linear regression (Hastie et al., 2022; Canatar et al., 2021; Belkin et al., 2019; Advani & Ganguli, 2016; Mel & Ganguli, 2021; Bartlett et al., 2020), but we provide a proof based on random projections and random matrix theory in Section D.5. This expression exhibits double descent behavior: in the overparameterized regime, the generalization error first decreases, then becomes infinite as $\gamma \to 1$, while in the underparameterized regime, the generalization error monotonically decreases with increasing $\gamma$. As we will see in Section 3.3, this double descent behavior leads to two distinct regions in the transferability phase diagram. The fact that scratch-trained performance can be arbitrarily bad is a result of the implicit regularization of gradient flow on this model. This effect can be eliminated by appropriately regularizing the scratch-trained model with weight decay. In the interest of analytic tractability we do not include regularization when training from scratch, but we explore its effects in simulation in Appendix G Fig. 5

## 3.3. Linear transfer

The simplest transfer learning method is known as linear transfer, in which only the final layer weights of the pretrained network are trained on the target task. In particular, $\{\boldsymbol{W}_l\}_{l \leq L-1}$ are fixed after pretraining and $\hat{\boldsymbol{W}}_L$ solves the linear regression problem with features $\boldsymbol{\Phi} = \boldsymbol{X} \boldsymbol{W}_1 \ldots \boldsymbol{W}_{L-1}$.

$$\hat{\boldsymbol{W}}_L = \arg \min_{\hat{\boldsymbol{W}}_L \in \mathbb{R}^d} \frac{1}{2n} \|\boldsymbol{\Phi} \boldsymbol{W}_L - \boldsymbol{y}_{\mathrm{t}}\|_2^2 \qquad (9)$$

When there are multiple solutions to the optimization problem (9), we choose the solution with minimum norm. We characterize the generalization error of linear transfer in the following theorem.

**Theorem 3.7.** *Under Assumptions 3.1 and 3.2, and assuming the source-target overlap is* $\theta$, *the expected generalization error of the linearly transferred model is an explicit function of* $\theta$, *the label noise* $\sigma$, *and the dataset size* $n$:

$$\mathbb{E}_{\mathcal{D}} \mathcal{R}_{\mathrm{lt}} = \sin^2 \theta + \frac{\sigma^2 + \sin^2 \theta}{n - 2}. \qquad (10)$$

We prove Theorem 3.7 in Appendix D.6. The structure of the result in Theorem 3.7 merits some discussion. After pretraining in the feature learning regime $\alpha \to 0$, the feature space of the network has sparsified so that it can only express functions along $\boldsymbol{\beta}_{\mathrm{s}}$ (Theorem 3.4). Since the features of the network cannot change in linear transfer, the

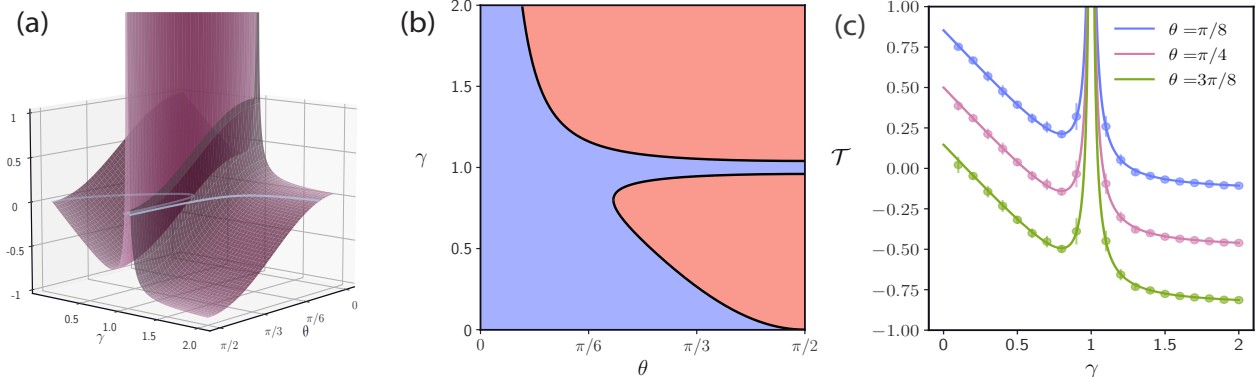

*Figure 1.* **Linear transferability phase diagram.** We pretrain a linear network (7) with $L = 2$ and $d = 500$ to produce labels from linear source function $y = \boldsymbol{\beta}_s^T \boldsymbol{x} + \epsilon$ using the population loss (2). We then retrain the final layer weights on a sample of $n = \gamma d$ points $(\boldsymbol{x}_i, y_i = \boldsymbol{\beta}_t^T \boldsymbol{x}_i + \epsilon_i)$ where $\boldsymbol{\beta}_s^T \boldsymbol{\beta}_t = \cos\theta$ and $\epsilon_i \sim \mathcal{N}(0, \sigma = 0.2)$, and compare its generalization error to that of a model trained from scratch on the target dataset. **(a)** The theoretical transferability surface (11) as a function of target dataset size $\gamma$ and task overlap $\theta$. Light blue lines indicate the boundary between positive and negative transfer. **(b)** Top-down view of Fig. 1(a) shaded by sign of transferability. Red regions indicate negative transfer $\mathcal{T} < 0$, blue region indicates positive transfer $\mathcal{T} > 0$. **(c)** Slices of the transferability surface (11) for constant $\theta$. Solid lines represent theoretical values, circles are points from experiments. Error bars represent the standard deviation over 20 draws of the target set.

main contribution to the generalization error is $\sin^2\theta$, which can be viewed as the norm of the projection of the target function into the space orthogonal to the features spanned by the pretrained network. This is an irreducible error that is the best case risk given that the features cannot change. The second term comes from the finiteness of the training set. Since linear transfer learns from a finite sample of training points, minimizing the training error can overfit the noise and the learned function distorts away from the ground truth. Luckily, since the pretrained feature space has sparsified, the effect of finite sampling and additive label noise decays as $\sim 1/n$, effectively filtering out the $d$-dimensional noise by projecting it onto a single vector. Compare this to the generalization of the scratch trained network (8). There, the features of the equivalent linear regression problem, $\boldsymbol{X}$, have support over all $d$-dimensions, so there is no irreducible error term. The expressivity, however, comes at a cost. Each dimension of the regression vector is vulnerable to noise in the training data, and the projection of the target function onto the feature space is strongly distorted due to finite sampling (i.e. $\sim \gamma$). We can precisely analyze this trade off by comparing (8) and (10). In the limit $n, d \to \infty$, the transferability (5) is

$$\mathcal{T}_{\text{lt}} = \begin{cases} \frac{(1-\gamma)^2 + \gamma\sigma^2}{1-\gamma} - \sin^2\theta & \gamma < 1 \\ \frac{\sigma^2}{\gamma - 1} - \sin^2\theta & \gamma > 1 \end{cases} \quad (11)$$

which is plotted in Fig. 1(a). From (11) we can identify the regions of negative transfer for this model, which are shaded in red in Fig. 1(b). In the underparameterized regime $(\gamma > 1)$, there is negative transfer for all $\gamma - 1 > \frac{\sigma^2}{\sin^2\theta}$. In words, at fixed $\gamma$ and $\sigma$, i.e., fixing the number of data points and label noise, transfer efficiency degrades as the norm of the out-of-subspace component increases.

In the overparameterized regime $(\gamma < 1)$, negative transfer only occurs when $\sigma < 1$. This can be viewed as a condition on the signal-to-noise ratio of the target data: $\text{SNR} = \|\boldsymbol{\beta}_t\|_2^2/\sigma^2 = 1/\sigma^2$. When $\text{SNR} < 1$, scratch training can never recover the underlying vector $\boldsymbol{\beta}_t$ and pretraining is always beneficial. When $\text{SNR} > 1$, negative transfer occurs when $\theta \in (\arccos(1-\sigma), \pi/2)$ and $\gamma \in (\gamma_+, \gamma_-)$ where $\gamma_\pm = \frac{1}{2}[(1 + \cos^2\theta - \sigma^2) \pm \sqrt{(1 + \cos^2\theta - \sigma^2)^2 - 4\cos^2\theta}]$. In the noiseless case $\sigma \to 0$, this expression simplifies to $\theta \in (0, \pi/2)$, $\cos^2\theta < \gamma < 1$ (see Appendix G Fig. 7). We can view $\gamma$ as a dimensionless measure of the amount of target data, and $\cos^2\theta$ as the amount of power the target function has in the subspace of the pretraining task. The condition for negative transfer is satisfied when there is more target data than there is power in the pretrained subspace. As $\sigma$ increases, the region of negative transfer shrinks, since the noise corrupts the scratch trained accuracy. Finally we mention that the two regions of negative transfer in Fig. 1 are separated by

positive transfer that persists even when $\theta = \pi/2$. We dub this effect *anomalous positive transfer*, since the pretrained features are completely orthogonal to those in the target, yet transferability is still positive. In this regime, transfer is positive soley because of the disproportionately large amount of data in the source task, not because pretraining learned useful features for the downstream task. By comparing the transferred model to a regularized scratch-trained model, we can eliminate this effect, which we show in simulation in Appendix G Fig. 5. In Appendix G Fig. 4 we demonstrate that distributional discrepency measures are indeed misleading: neither $D_{\mathrm{KL}}$, nor $W_1$ are negatively correlated with increased transferability.

### 3.3.1. RIDGE REGULARIZATION CANNOT FIX NEGATIVE TRANSFER

In the previous section, the network sparsified to features that incompletely described the target function, leading to negative transfer given sufficient target data. A common approach to mitigate this kind of multicollinearity in linear regression is to add an $\ell_2$ penalty to the regression objective (9) so that

$$\hat{\boldsymbol{W}}_L = \arg\min_{\hat{\boldsymbol{W}}_L \in \mathbb{R}^d} \frac{1}{2n} \|\boldsymbol{\Phi}\boldsymbol{W}_L - \boldsymbol{y}_{\mathrm{t}}\|_2^2 + \frac{\lambda}{2}\|\boldsymbol{W}_L\|_2^2. \quad (12)$$

In the following theorem, which we prove in Appendix D.7, we show that the generalization error for regularized linear transfer is a strictly increasing function of the ridge parameter $\lambda$, leading to a larger region of negative transfer for any $\lambda > 0$ (Fig. 6).

**Theorem 3.8.** *Under Assumptions 3.1 and 3.2, and assuming the source-target overlap is $\theta$, the expected generalization error of the ridge linear transfer model over the training data is*

$$\lim_{n \to \infty} \mathbb{E}_{\mathcal{D}} \mathcal{R}_{\mathrm{lt}}^{\lambda} = 1 - \frac{(1 + 2\lambda)}{(1 + \lambda)^2} \cos^2\theta \quad (13)$$

Ridge regression attenuates the power of the predictor in all directions of the data, including the direction parallel to the signal. Due to sparisification of Theorem 3.4, $\ell_2$ regularization is non-optimal and hence regularization impairs generalization by attenuating useful features, i.e., those with $\theta < \pi/2$.

### 3.4. Fine-tuning

Another common transfer learning strategy is *fine-tuning*, in which all model parameters are trained on the target task from the pre-trained initialization. For general nonlinear models, analyzing the limit points of gradient flow from arbitrary initialization is a notoriously difficult task. For the deep linear model however, we can solve for the expected generalization error of fine-tuning exactly.

**Theorem 3.9.** *Under Assumptions 3.1 and 3.2, and assuming the source-target overlap is $\theta$, the expected generalization error of the fine-tuned model over the training data is:*

$$\mathbb{E}_{\mathcal{D}} \mathcal{R}_{\mathrm{ft}} = \begin{cases} \mathbb{E}_{\mathcal{D}} \mathcal{R}_{\mathrm{sc}} + (1-\gamma)(1 - 2\cos\theta) & \gamma \leq 1 \\ \mathbb{E}_{\mathcal{D}} \mathcal{R}_{\mathrm{sc}} & \gamma > 1 \end{cases} \quad (14)$$

*where $\mathbb{E}_{\mathcal{D}} \mathcal{R}_{\mathrm{sc}}$ is the expected generalization error of the scratch trained model*

Theorem 3.9 is proven in Appendix D.8. Theorem 3.9 yields an expression for the fine-tuning transferability, which is plotted in Fig. 2(a):

$$\mathcal{T}_{\mathrm{ft}} = \begin{cases} (\gamma - 1)(1 - 2\cos\theta) & \gamma \leq 1 \\ 0 & \gamma > 1 \end{cases} \quad (15)$$

When the model is underparameterized $\gamma > 1$, there is a unique global minimum in the space of $\boldsymbol{\beta} = \boldsymbol{W}_1 \boldsymbol{W}_2 \cdots \boldsymbol{W}_L$. Since gradient flow converges to a global minimum, (Theorem 3.5), fine tuning loses the memory of the pretrained initialization leading to zero transferability (white region in Fig. 2(b)). When the network is overparameterized, however, there is a subspace of global minima. We show in the Section D.8 that the pretrained initialization induces an implicit bias of gradient flow away from the minimum norm solution. When the target vector has the majority of its norm along the pretraining subspace, i.e. $\cos\theta > 1/2$, the pretrained features are beneficial, leading to positive transfer. For $\cos\theta < 1/2$, however, the pretrained features bias the network too strongly toward the source task, leading to negative transfer.

## 4. Student-teacher ReLU networks

In the following, we demonstrate that many of the results from our analytically solvable model also hold, qualitatively, in the more complicated setting of linear transfer with nonlinear networks. In particular, we choose a model of the form $f(\boldsymbol{x}) = \frac{1}{m} \sum_{i=1}^{m} c_i \sigma(\boldsymbol{w}_i^T \boldsymbol{x})$ where $\sigma(y) = \max\{0, y\}$ is the ReLU activation. We scale the model by $1/m$ to place the network in the mean field, feature learning regime (Chizat et al., 2019; Mei et al., 2018; Rotskoff & Vanden-Eijnden, 2022). As in the deep linear model, we choose source and target functions that are representable by the network. That is, we study this model in the student teacher setting. To vary the level of feature space overlap between the source and target functions, we define a network of $m_*$ neurons for the target task, and generate the source network by ablating a fraction $\mu$ of the hidden neurons form the target. More precisely, let $\mathcal{A}$ be a uniformly random subset of the index set $\{1, 2, \cdots m_*\}$ with

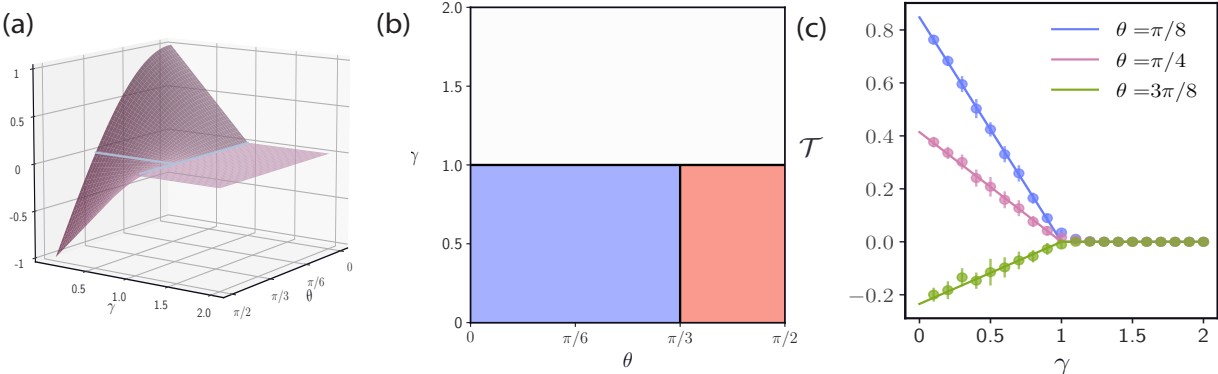

*Figure 2.* **Fine-tuning transferability surface** Using the same transfer setup as in Fig. 1 we fine tune all of the weights on the target dataset starting from the pretrained weight initialization. **(a)** The theoretical transferability surface (15) as a function of target dataset size $\gamma$ and task overlap $\theta$. The light blue line parallel to the $\gamma$ axis indicates the boundary between positive and negative transfer, while the one parallel to the $\theta$ axis indicates the boundary for zero transferability. **(b)** Top-down view of Fig. 2(a) shaded by sign of transferability. Red region indicates negative transfer $\mathcal{T} < 0$, blue region indicates positive transfer $\mathcal{T} > 0$. The white region indicates no transfer benefit $\mathcal{T} = 0$. **(c)** Slices of the transferability surface (15) for constant $\theta$. Solid lines represent theoretical values, circles are points from experiments. Error bars represent the standard deviation over 20 draws of the target set.

$|\mathcal{A}| = \mu m_*$. Then

$$f_s^*(\boldsymbol{x}) = \frac{1}{(1-\mu)m_*} \sum_{i \in \mathcal{A}^c} c_i^* \sigma(\boldsymbol{w}_i^{*T}\boldsymbol{x}) \qquad (16)$$

$$f_t^*(\boldsymbol{x}) = \frac{1}{m_*} \sum_{i=1}^{m_*} c_i^* \sigma(\boldsymbol{w}_i^{*T}\boldsymbol{x}) \qquad (17)$$

Thus the source has $\mu m^*$ *fewer* hidden features than the target task, and so the fraction $\mu$ controls the degree of discrepancy between source and target feature spaces. In essence when $\mu = 0$ the source and target spaces are identical. However, as $\mu$ increases, an increasing fraction of new target features, that were not present in pre-training, must be learned. We constrain the hidden features in the model, source, and target to the $d$-dimensional unit sphere $\boldsymbol{w}_i, \boldsymbol{w}_i^* \in \mathbb{S}^{d-1}$. As in the deep linear model, we choose $\boldsymbol{x} \sim \mathcal{N}(0, \boldsymbol{I}_d)$, train the source task on the population loss, which can be computed exactly for this model, and the target task on a finite sample of $n$ data points.

Previous work (Rotskoff & Vanden-Eijnden, 2022; Mei et al., 2018; Chizat, 2020) has shown that in the over-parameterized setting $m \gg m_*$, gradient flow will converge to a global minimizer of the population loss, so that $\lim_{m \to \infty} \lim_{t \to \infty} f(\boldsymbol{x}) = f_s^*(\boldsymbol{x})$, which establishes that the trained network builds a representation of $f_s^*(\boldsymbol{x})$ in the mean field limit. This does not necessarily mean that all of the hidden neurons of the model converge to those of

the teacher, since any superfluous weight directions can be eliminated by setting the corresponding output weight to zero. However, we demonstrate empirically in Fig. 3(a)-(b) that this relationship is preserved at the level of the model's kernel, so that $k(\boldsymbol{x}, \boldsymbol{x}') = \frac{1}{m} \sum_{i=1}^{m} \sigma(\boldsymbol{w}_i^T\boldsymbol{x})\sigma(\boldsymbol{w}_i^T\boldsymbol{x}') \approx \frac{1}{(1-\mu)m_*} \sum_{i \in \mathcal{A}^c} \sigma(\boldsymbol{w}_i^{*T}\boldsymbol{x})\sigma(\boldsymbol{w}_i^{*T}\boldsymbol{x}')$. This observation is analogous to Theorem 3.4: training in the feature learning regime causes the model's features to sparsify to those present in the target function.

Now, linear transfer in this model can be formulated as a kernel interpolation problem with this kernel. The generalization error of kernel interpolation can be separated into an $n$-dependent component, and an irreducible error term which corresponds to the norm of the projection of the target function into the subspace of $L_2(p)$ orthogonal to the RKHS defined by the kernel:

$$\mathbb{E}_D \mathcal{R}_{lt} = C(n) + \|P^\perp f_t^*(\boldsymbol{x})\|_{L_2}^2. \qquad (18)$$

As expected, the norm of this projection increases monotonically with $\mu$ as shown in Fig. 3(d). We show how to compute this projection in Appendix E. In the deep linear setting, $\|P^\perp f_t^*(\boldsymbol{x})\|_{L_2}^2 = \sin^2\theta$, and $C(n) \sim 1/n$. While the asymptotic, typical generalization error of kernel regression has been studied in (Canatar et al., 2021), for the purposes of estimating the generalization error of the transferred model, we assume here that this generalization error is dominated by this irreducible term for the large $n$ target

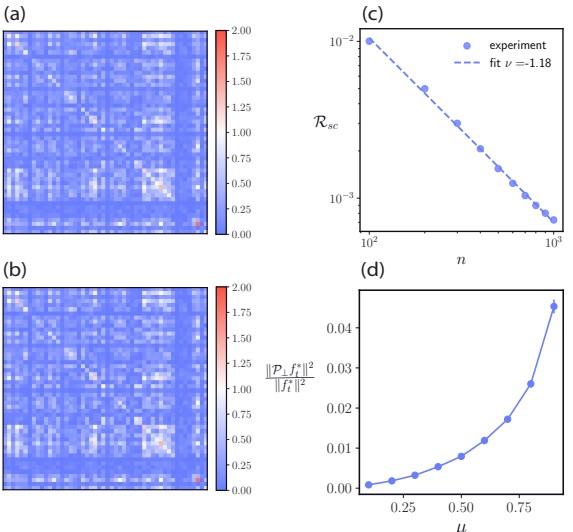

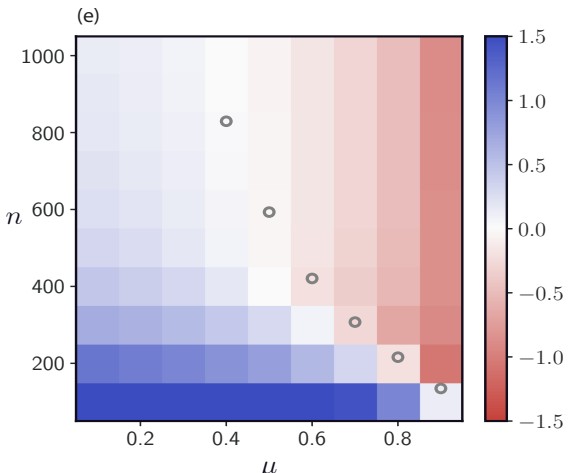

*Figure 3.* **Linear transfer in two-layer ReLU networks** We train a two layer ReLU network with $m = 1000$ neurons on a teacher with $m_* = 100$ neurons and $d = 100$ dimensional gaussian data, according to the ablated transfer setup (16), (17). For these experiments, we set the label noise $\sigma = 0$. **(a)** Gram matrix from the kernel of the pretrained model **(b)** Gram matrix from the kernel of the ground truth source function $f_s^*(\boldsymbol{x})$. The two gram matrices are nearly indistinguishable suggesting that the kernel sparsifies to the represent features in the source task. **(c)** Generalization error of the scratch trained model as a function of dataset size $n$, fit to a power law **(d)** Norm of out-of-RKHS component of target function $\|P^\perp f_t^*(\boldsymbol{x})\|_{L_2}^2$, normalized by target function norm $\|f_t^*(\boldsymbol{x})\|_{L_2}^2$ as a function of excess target features $\mu$. **(e)** Heat map of transferability as a function of excess target features $\mu$ and dataset size $n$. We normalize the transferability by variance in the target data. Gray circles represent the point of negative transfer predicted by our theory. Results are averaged over 100 realizations of the data and 10 realizations of random draws of the teacher.

dataset sizes we consider, just as we showed for the deep linear model.

However, an expression for the generalization error of the scratch-trained model is also needed to derive the transferability. We are not aware of a theory of generalization error for infinite width nonlinear networks trained on a finite data in the mean field regime. Intriguingly, however, we demonstrate empirically (Fig. 3(c)) that the generalization error obeys a power law $\mathcal{R}_{\text{sc}} \sim An^{-\nu}$ with $\nu = 1.18$. By setting our theoretically predicted generalization error of our transferred model $\|P^\perp f_t^*(\boldsymbol{x})\|_{L_2}^2$ equal to the empirically observed scaling law $An^{-\nu}$ for our scratch-trained model, we can approximately identify the point of negative transfer in $n$ for any given $\mu$ (gray circles in Fig. 3(e)). It is clear from Fig. 3(e) that this heuristic for finding the boundary between positive and negative transfer becomes more accurate as the number of target points becomes large, since the $n$-dependent component of the kernel regression generalization error goes to zero in this limit. The phase diagram in Fig. 3 for noiseless ReLU networks resembles the phase diagram for linear transfer with deep linear networks in the noiseless setting with $\sigma = 0$ (Appendix G Fig.7. 8).

Overall, this demonstrates that we are able to predict the phase boundary between positive and negative transfer in the ReLU case, using our conceptual understanding in the deep linear case.

## 5. Conclusion

In this paper, we highlight the importance of thinking about transfer learning in the context of the feature space of the pretrained model. We rigorously identify the number of data points necessary for transfer learning to outperform scratch training as a function of feature space overlap in deep linear networks. We also demonstrate that our understanding of linear transfer carries over to shallow nonlinear networks as well. One of our primary findings is that transferability is inherited from the learned features of the pretraining task. In the rich training regime, this can lead to an inability for the pretrained model to transfer to tasks outside the source feature space. On the other hand, a model trained in the lazy regime is unlikely to outperfrom scratch training, since features are not updated in this limit. This suggests that models trained somewhere along the lazy-to-rich hier-

archy may be more flexible in their transfer capabilities. In Appendix G Fig. 9 we generate a sweep of nonlinear models trained with varying degrees of feature learning on the source task and show that we can eliminate negative transfer if the pretrained model lies optimally between the lazy and rich regimes. These experiments demonstrate that regularizing pretrained models to avoid feature sparsification in the source task is a promising direction for improving transfer learning capabilities. With this work, we hope to advance the idea that transfer learning performance should be understood through the learned feature space of the pretrained model and not as a property of the dataset alone.

## Impact Statement

This work aims to understand the theoretical aspects of transfer learning, and advances the idea that the feature space of the pretrained model determines the capability of the model to transfer to downstream tasks. This theoretical understanding could be used to create data and compute efficient pretraining procedures that maximize transferability to downstream tasks of interest.

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

## A. Dataset similarity is not predictive of transfer efficiency

The common wisdom in transfer learning is that related tasks should transfer effectively to one another. While it may seem that closeness of task distributions should correlate with transfer performance, we show that this is not necessarily the case. In particular, we select a member of each family and prove that, within our model, one can achieve positive transfer ($\mathcal{T} > 0$) with distributions that are arbitrarily far apart. Two functions representable with the same features can be "far apart". We formalize this notion with the following theorem.

**Assumption A.1.** We assume $f \in L_2(\mathbb{R}^d, p)$ and for each $\boldsymbol{x} \in \mathbb{R}^d$ we define the random variable $y : \mathbb{R}^d \to \mathbb{R}$ through the relation $y = f(\boldsymbol{x}) + \epsilon$ with $\epsilon \sim \mathcal{N}(0, \sigma^2)$. Let $p_f(\boldsymbol{x}, y)$ denote the joint probability density of $\boldsymbol{x}$ and $y$. We assume $\Phi \subset L_2(p)$ is a linear subspace with orthonormal basis $\{\phi_i\}_{i=1}^M$ and $M$ may be infinite.

**Theorem A.2.** *Assume A.1. Then for any $f \in \Phi$, and any $\delta > 0$ there exists $g \in \Phi$ such that*

$$\gamma_\beta(p_f, p_g) \geq \delta$$

*where $\gamma_\beta(p, p')$ is the Dudley Metric. Similarly, for any $f \in \Phi$, and any $\delta > 0$ there exists $g \in \Phi$ such that*

$$D_{\mathrm{KL}}(p_f \| p_g) \geq \delta$$

*where $D_{\mathrm{KL}}(p_f \| p_g)$ is the Kullback Leibler divergence.*

We prove this theorem in Appendix D.1. We note that this theorem also holds for any IPM over a function class that is larger than the class of Bounded Lipschitz functions. In particular, the theorem holds for the Monge-Kantorovich ($W_1$) metric, since any function that satisfies $\|f\|_{\mathrm{BL}} \leq 1$ also satisfies $\|f\|_L \leq 1$.

Theorem A.2 demonstrates that for a given source distribution, one can always find a target distribution generated from the same feature space that is *arbitrarily* distant with respect to these metrics, perhaps creating the illusion that transfer is likely to fail. However, even when the distance is large, if the source and target functions lie in the *same* feature space and pretraining creates a basis for this space, transfer to the target task will be positive, since only the output weights need to be relearned in the target task. We show this is indeed the case for deep linear networks in the following section.

## B. Initialization assumption

Following (Yun et al., 2021) we place the following constraint on the initialization for some $\lambda > 0$.

$$\bar{\boldsymbol{W}}_l^T \bar{\boldsymbol{W}}_l - \bar{\boldsymbol{W}}_{l+1} \bar{\boldsymbol{W}}_{l+1}^T \succcurlyeq \lambda I \tag{19}$$

To our knowledge, this is the most general assumption on weight initializations in the literature that leads to the implicit biases that are crucial for our analysis. This initialization scheme generalizes that in Wu et al. (2019); Atanasov et al. (2021).

## C. Fine tuning transferability with finite source data

In the main text, we consider training the source task on the population loss, which can be understood as having infinite data points in the source task. Although the number of source data points typically dominates the number of target data points in real-world applications, it is interesting to consider the scenario in which the dataset sizes are comparable. To this end, we consider a source task with $n_s = \gamma_s d$ data points and a target task with $n_t = \gamma_t d$ data points, generalizing the setup of 3. Since the source dataset is now finite, we also add label noise to the source task so that the labels are generated as

$$y_s = \boldsymbol{\beta}_{\mathrm{s}}^T \boldsymbol{x} + \epsilon_s \tag{20}$$

$$y_t = \boldsymbol{\beta}_{\mathrm{t}}^T \boldsymbol{x} + \epsilon_t \tag{21}$$

where $\epsilon_s \sim \mathcal{N}(0, \sigma_s^2)$ and $\epsilon_t \sim \mathcal{N}(0, \sigma_t^2)$. Aside from these changes, the setup in the following is the same as that in 3. We derive the following expression for the expected generalization error using full fine-tuning, which we prove in Appendix D.9.

**Theorem C.1.** *Under assumptions above, and assuming the source-target overlap is $\theta$, the expected transferability of the fine-tuned model over the training data is:*

$$\mathcal{T}_{\text{ft}} = \mathbb{E}_{\mathcal{D}_t} \mathcal{R}_{\text{sc}} - \mathbb{E}_{\mathcal{D}_s, \mathcal{D}_t} \mathcal{R}_{\text{ft}} = \begin{cases} \frac{\gamma_t - 1}{1 - \gamma_s} \sigma_s^2 \gamma_s + \gamma_s(\gamma_t - 1)(1 - 2\cos\theta) & \gamma_s, \gamma_t \leq 1 \\ (\gamma_t - 1)(1 - 2\cos\theta) - \frac{\gamma_t - 1}{1 - \gamma_s} \sigma_s^2 & \gamma_s > 1 > \gamma_t \\ 0 & \gamma_t > 1 \end{cases} \tag{22}$$

Note that this expression agrees with (15) in the limit $\gamma_s \to \infty$ and that the negative transfer boundary is completely determined by $\gamma_s$, since $\gamma_t$ only enters the expression via a global multiplicative factor. Secondly, without label noise in the source task, the phase diagram is the same as in Fig 2. That is to say, fine tuning does not depend on the amount of source data if there is no label noise. In the general case, for fixed $\gamma_s$, the boundary between positive and negative transferability occurs at a fixed value of $\theta$:

$$\cos\theta_* = \begin{cases} \frac{1}{2} \left( \frac{\sigma_s^2 + 1 - \gamma_s}{1 - \gamma_s} \right) & \gamma_s < 1 \\ \frac{1}{2} \left( \frac{\sigma_s^2 + \gamma_s - 1}{\gamma_s - 1} \right) & \gamma_s > 1 \end{cases} \tag{23}$$

Therefore, the fine tuning transferability phase diagram with a finite source dataset is qualitatively the same as that with infinite source data (Fig. 2), but the position of the phase boundary is modulated by the size of the source dataset.

## D. Proofs

### D.1. Proof of Theorem A.2

We begin by recalling the definition of the Dudley Metric

$$\gamma_\beta(p, q) = \sup_{\|h\|_{\text{BL}} \leq 1} |\mathbb{E}_p h - \mathbb{E}_q h| \tag{24}$$

$$\|h\|_{\text{BL}} = \|h\|_L + \|h\|_\infty \tag{25}$$

By conditioning $p_f(x, y)$ and $p_g(x, y)$ on $x$, we can write

$$\gamma_\beta(p_f, p_g) = \sup_{\|h\|_{BL} \leq 1} \left| \frac{1}{\sqrt{2\pi\sigma^2}} \int \left[ h(x, y) e^{\frac{-(y - f(x))^2}{2\sigma^2}} - h(x, y) e^{\frac{-(y - g(x))^2}{2\sigma^2}} \right] p(x) \mathrm{d}x \mathrm{d}y \right| \tag{26}$$

$$\geq \left| \frac{1}{\sqrt{2\pi\sigma^2}} \int \left[ \frac{\cos(y)}{2} e^{\frac{-(y - f(x))^2}{2\sigma^2}} - \frac{\cos(y)}{2} e^{\frac{-(y - g(x))^2}{2\sigma^2}} \right] p(x) \mathrm{d}x \mathrm{d}y \right| \tag{27}$$

$$= \left| \frac{e^{-\sigma^2/2}}{2} \int \left[ \cos(f(x)) - \cos(g(x)) \right] p(x) \mathrm{d}x \right| \tag{28}$$

$$\geq \frac{e^{-\sigma^2/2}}{2} \int \left[ f(x)^2 + g(x)^2 \right] p(x) \mathrm{d}x \tag{29}$$

$$\tag{30}$$

The first inequality follows from the fact that $\|\frac{\cos(y)}{2}\|_{\text{BL}} = 1$, and the second follows from the identity $\cos(x) + x^2 \geq \cos(z) - z^2$ for any $x, z \in \mathbb{R}$. We can expand $f$ in the orthonormal basis $\{\phi_i\}_{i=1}^M$ as $f = \sum_{i=1}^M \alpha_i \phi_i$, so that

$$\int f(x)^2 p(x) \mathrm{d}x = \sum_{i,j} \alpha_i \alpha_j \int p(x) \phi_i(x) \phi_j(x) \mathrm{d}x = \sum_i \alpha_i^2 \tag{31}$$

Since, $f \in L_2(p)$, the sum on right hand side of (31) converges to some $a < \infty$. We can choose $g = \sqrt{\left| \frac{2\delta e^{\sigma^2/2} - a}{a} \right|} \sum_{i=1}^M \alpha_i \phi_i$ which completes the first half of the proof. To prove the result about the KL divergence,

can directly calculate $\mathcal{D}_{KL}(p_f\|p_g)$

$$\mathcal{D}_{KL}(p_f\|p_g) = \frac{1}{\sqrt{2\pi\sigma^2}} \int p(x)e^{-\frac{(y-f(x))^2}{2\sigma^2}} \left[\frac{(y-g(x))^2}{2\sigma^2} - \frac{(y-f(x))^2}{2\sigma^2}\right] \mathrm{d}x\mathrm{d}y \tag{32}$$

$$= \frac{1}{\sqrt{2\pi\sigma^2}} \int p(x)e^{-\frac{(y-f(x))^2}{2\sigma^2}} \left[g(x)^2 - f(x)^2 - 2yg(x) + 2yf(x)\right] \mathrm{d}x\mathrm{d}y \tag{33}$$

$$= \frac{1}{2\sigma^2} \left[\|f\|_{L_2}^2 + \|g\|_{L_2}^2 - 2\langle f, g\rangle\right] \tag{34}$$

$$= \frac{1}{2\sigma^2} \|f - g\|_{L_2(p)}^2 \tag{35}$$

For any $\delta > 0$ we can choose $g = -\alpha f$ with $\alpha > \frac{\sigma\delta^{1/2}}{\|f\|_{L_2(p)}}$ which completes the proof.

### D.2. Proof of Lemma 3.3

We proceed by bounding the dynamics of the loss by an exponentially decaying dynamics, proving convergence to a global minimum. Then we show that the value of $\boldsymbol{\beta}$ at a global minimum is unique. To begin, note that the matrix

$$\boldsymbol{D}_l = \boldsymbol{W}_l^T \boldsymbol{W}_l - \boldsymbol{W}_{l+1}\boldsymbol{W}_{l+1}^T \tag{36}$$

is an invariance of the gradient flow dynamics, so that $\boldsymbol{D}(t) = \boldsymbol{D}(0) = \alpha^2(\bar{\boldsymbol{W}}_l^T\bar{\boldsymbol{W}}_l - \bar{\boldsymbol{W}}_{l+1}\bar{\boldsymbol{W}}_{l+1}^T)$ for all time (Atanasov et al., 2021; Kunin et al., 2024; Yun et al., 2021). Let $\boldsymbol{r} = (\boldsymbol{W}_1\boldsymbol{W}_2\ldots\boldsymbol{W}_L - \boldsymbol{\beta}_\mathrm{s})$ and note that

$$\dot{\mathcal{L}} = \sum_{l=1}^{L}\langle\nabla_l\mathcal{L}, \dot{\boldsymbol{W}}_l\rangle \tag{37}$$

$$= -\sum_{l=1}^{L}\|\nabla_l\mathcal{L}\|_F^2 \tag{38}$$

$$\leq \|\nabla_L\mathcal{L}\|_F^2 \tag{39}$$

$$= -\|\boldsymbol{W}_{L-1}^T\ldots\boldsymbol{W}_1^T\boldsymbol{r}\|_2^2 \tag{40}$$

$$\leq -2\sigma_{\min}^2(\boldsymbol{W}_{L-1}^T\ldots\boldsymbol{W}_1^T)\mathcal{L} \tag{41}$$

$$\tag{42}$$

where $\sigma_{\min}(\boldsymbol{W}_{L-1}^T\ldots\boldsymbol{W}_1^T)$ is the smallest singular value of $\boldsymbol{W}_{L-1}^T\ldots\boldsymbol{W}_1^T$. To proceed we bound $\sigma_{\min}(\boldsymbol{W}_{L-1}^T\ldots\boldsymbol{W}_1^T)$ away from zero by showing that $\boldsymbol{W}_{L-1}\ldots\boldsymbol{W}_1\boldsymbol{W}_1^T\ldots\boldsymbol{W}_{L-1}^T$ is positive definite

$$\boldsymbol{W}_{L-1}^T\ldots\boldsymbol{W}_1^T\boldsymbol{W}_1\ldots\boldsymbol{W}_{L-1} = \boldsymbol{W}_{L-1}^T\ldots\boldsymbol{W}_2^T(\boldsymbol{W}_2\boldsymbol{W}_2^T + \boldsymbol{D}_1)\boldsymbol{W}_2\ldots\boldsymbol{W}_{L-1} \tag{43}$$

$$\succcurlyeq \boldsymbol{W}_{L-1}^T\ldots\boldsymbol{W}_3^T(\boldsymbol{W}_2^T\boldsymbol{W}_2)^2\boldsymbol{W}_3\ldots\boldsymbol{W}_{L-1} \tag{44}$$

$$\vdots$$

$$\succcurlyeq (\boldsymbol{W}_{L-1}^T\boldsymbol{W}_{L-1})^{L-1} \tag{45}$$

$$= (\boldsymbol{W}_L\boldsymbol{W}_L^T + \boldsymbol{D}_L)^{L-1} \tag{46}$$

$$\succcurlyeq (\alpha^2\lambda)^{L-1} \tag{47}$$

where we have used the conservation law (36) and the initialization assumption (19). We now have

$$\dot{\mathcal{L}} \leq -2(\alpha^2\lambda)^{L-1}\mathcal{L} \tag{48}$$

$$\implies \mathcal{L}(t) \leq \mathcal{L}(0)e^{-2(\alpha^2\lambda)^{L-1}t} \tag{49}$$

$$\implies \lim_{t\to\infty}\mathcal{L}(t) = 0 \tag{50}$$

Since the loss converges to zero, $\lim_{t\to\infty}\boldsymbol{W}_1\boldsymbol{W}_2\ldots\boldsymbol{W}_L = \lim_{t\to\infty}\boldsymbol{\beta} = \boldsymbol{\beta}_\mathrm{s}$, which is unique. Note that while this solution is unique in function space, it is degenerate in parameter space.

### D.3. Proof of Theorem 3.4

To prove the feature space sparsification, we rely on the following Lemma, which is proven in (Yun et al., 2021) (see Section H.2). So that this work is self-contained, we include the proof here.

**Lemma D.1.** *Under gradient flow on the population objective (2) or the empirical objective (3),*

$$\boldsymbol{W}_l = \sigma_l(t)\boldsymbol{u}_l(t)\boldsymbol{v}_l(t) + \mathcal{O}(\alpha^2) \tag{51}$$

*for all time. Furthermore*

$$\lim_{\alpha \to 0} \lim_{t \to \infty} (\boldsymbol{u}_{l+1}(t)^T \boldsymbol{v}_l(t))^2 = 1 \tag{52}$$

*Proof.* To prove Lemma D.1 we bound the difference $\|\boldsymbol{W}_l\|_F^2 - \|\boldsymbol{W}_l\|_{op}^2$ which is equal to the norm of the subleading singular vectors of $\boldsymbol{W}_l$ and show that this bound is proportional to $\alpha^2$. The argument here follows that in ((Yun et al., 2021)). Taking the trace of both sides in (36) we have

$$\|\boldsymbol{W}_l\|_F^2 - \|\boldsymbol{W}_{l+1}\|_F^2 = \alpha^2(\|\bar{\boldsymbol{W}}_l\|_F^2 - \|\bar{\boldsymbol{W}}_{l+1}\|_F^2) \tag{53}$$

$$\sum_{k=l}^{L-1} \|\boldsymbol{W}_k\|_F^2 - \|\boldsymbol{W}_{k+1}\|_F^2 = \alpha^2 \sum_{k=l}^{L-1} (\|\bar{\boldsymbol{W}}_k\|_F^2 - \|\bar{\boldsymbol{W}}_{k+1}\|_F^2) \tag{54}$$

$$\|\boldsymbol{W}_l\|_F^2 - \|\boldsymbol{W}_L\|_F^2 = \alpha^2(\|\bar{\boldsymbol{W}}_l\|_F^2 - \|\bar{\boldsymbol{W}}_L\|_F^2) \tag{55}$$

Let $\boldsymbol{u}_l, \boldsymbol{v}_l$ be the top left and right singular vectors of $\boldsymbol{W}_l$. To bound the maximum singular value of $\boldsymbol{W}_l$ we have

$$\|\boldsymbol{W}_l\|_{op}^2 = \boldsymbol{v}_l^T \boldsymbol{W}_l^T \boldsymbol{W}_l \boldsymbol{v}_l \geq \boldsymbol{u}_{l+1}^T \boldsymbol{W}_l^T \boldsymbol{W}_l \boldsymbol{u}_{l+1} \tag{56}$$

$$= \boldsymbol{u}_{l+1}^T (\boldsymbol{D}_l + \boldsymbol{W}_{l+1}^T \boldsymbol{W}_{l+1}) \boldsymbol{u}_{l+1} \tag{57}$$

$$= \|\boldsymbol{W}_{l+1}\|_{op}^2 + \alpha^2 \boldsymbol{u}_{l+1}^T (\bar{\boldsymbol{W}}_l^T \bar{\boldsymbol{W}}_l - \bar{\boldsymbol{W}}_{l+1} \bar{\boldsymbol{W}}_{l+1}^T) \boldsymbol{u}_{l+1} \tag{58}$$

$$\geq \|\boldsymbol{W}_{l+1}\|_{op}^2 + \alpha^2(\|\bar{\boldsymbol{W}}_{l+1}\|_{op}^2 - \|\bar{\boldsymbol{W}}_l\|_{op}^2) \tag{59}$$

Summing this inequality from $l$ to $L-1$ we have

$$\|\boldsymbol{W}_l\|_{op}^2 \geq \|\bar{\boldsymbol{W}}_L\|_{op}^2 + \alpha^2(\|\bar{\boldsymbol{W}}_L\|_{op}^2 - \|\bar{\boldsymbol{W}}_l\|_{op}^2) \tag{60}$$

Combining (54) and (60) we have

$$\|\boldsymbol{W}_l\|_F^2 - \|\boldsymbol{W}_l\|_{op}^2 \leq \alpha^2(\|\bar{\boldsymbol{W}}_l\|_F^2 - \|\bar{\boldsymbol{W}}_L\|_F^2 + \|\bar{\boldsymbol{W}}_l\|_{op}^2 - \|\bar{\boldsymbol{W}}_L\|_{op}^2) \tag{61}$$

This shows all of the parameter matrices are approximately rank one with corrections upper bounded by $\mathcal{O}(\alpha^2)$, proving the first claim. To show the alignment of adjacent singular vectors we again take advantage of the invariant quantity (36)

$$\boldsymbol{v}_l^T \boldsymbol{W}_{l+1} \boldsymbol{W}_{l+1}^T \boldsymbol{v}_l = \boldsymbol{v}_l^T \boldsymbol{W}_l^T \boldsymbol{W}_l \boldsymbol{v}_l - \boldsymbol{v}_l^T \boldsymbol{D}_l \boldsymbol{v}_l \tag{62}$$

$$\geq s_l^2 - \alpha^2 \|\bar{\boldsymbol{W}}_l^T \bar{\boldsymbol{W}}_l - \bar{\boldsymbol{W}}_{l+1} \bar{\boldsymbol{W}}_{l+1}^T\|_{op}^2 \tag{63}$$

we also derive the following upper bound on (63)

$$\boldsymbol{v}_l^T \boldsymbol{W}_{l+1} \boldsymbol{W}_{l+1}^T \boldsymbol{v}_l = \boldsymbol{v}_l^T (s_{l+1}^2 \boldsymbol{u}_{l+1} \boldsymbol{u}_{l+1}^T \boldsymbol{W}_{l+1} \boldsymbol{W}_{l+1}^T - s_{l+1}^2 \boldsymbol{u}_{l+1} \boldsymbol{u}_{l+1}^T) \boldsymbol{v}_l \tag{64}$$

$$\leq s_{l+1}^2 (\boldsymbol{v}_l^T \boldsymbol{u}_{l+1})^2 + \|\boldsymbol{W}_{l+1}\|_F^2 - \|\boldsymbol{W}_{l+1}\|_F^2 \tag{65}$$

combining these two bounds

$$s_l^2 \leq s_{l+1}^2 (\boldsymbol{v}_l^T \boldsymbol{u}_{l+1})^2 + \alpha^2 \|\bar{\boldsymbol{W}}_l^T \bar{\boldsymbol{W}}_l - \bar{\boldsymbol{W}}_{l+1} \bar{\boldsymbol{W}}_{l+1}^T\|_{op}^2 + \|\boldsymbol{W}_{l+1}\|_F^2 - \|\boldsymbol{W}_{l+1}\|_F^2 \tag{66}$$

$$\leq s_{l+1}^2 (\boldsymbol{v}_l^T \boldsymbol{u}_{l+1})^2 + \alpha^2 \|\bar{\boldsymbol{W}}_l^T \bar{\boldsymbol{W}}_l - \bar{\boldsymbol{W}}_{l+1} \bar{\boldsymbol{W}}_{l+1}^T\|_{op}^2 + \alpha^2(\|\bar{\boldsymbol{W}}_l\|_F^2 - \|\bar{\boldsymbol{W}}_L\|_F^2 + \|\bar{\boldsymbol{W}}_l\|_{op}^2 - \|\bar{\boldsymbol{W}}_L\|_{op}^2) \tag{67}$$

where we have used the result derived in the previous proof for the second inequality. Finally, we derive an upper bound on this quantity

$$s_l^2 \geq \boldsymbol{u}_{l+1}^T \boldsymbol{W}_l^T \boldsymbol{W}_l \boldsymbol{u}_{l+1} \tag{68}$$

$$\geq s_{l+1}^2 - \alpha^2 \|\bar{\boldsymbol{W}}_l^T \bar{\boldsymbol{W}}_l - \bar{\boldsymbol{W}}_{l+1} \bar{\boldsymbol{W}}_{l+1}^T\|_{\text{op}}^2 \tag{69}$$

We can combine the upper and lower bounds and divide by $s_{l+1}^2$ to conclude

$$(\boldsymbol{v}_l^T \boldsymbol{u}_{l+1})^2 \geq 1 - \alpha^2 \frac{C_l}{s_{l+1}^2} \tag{70}$$

$$C_l = 2\|\bar{\boldsymbol{W}}_l^T \bar{\boldsymbol{W}}_l - \bar{\boldsymbol{W}}_{l+1} \bar{\boldsymbol{W}}_{l+1}^T\|_{\text{op}}^2 + \|\bar{\boldsymbol{W}}_l\|_F^2 - \|\bar{\boldsymbol{W}}_L\|_F^2 + \|\bar{\boldsymbol{W}}_l\|_{\text{op}}^2 - \|\bar{\boldsymbol{W}}_L\|_{\text{op}}^2 \tag{71}$$

This proves that adjacent singular vectors align as long as the singular values are bounded away from zero. To show that this requirement is satisfied at the end of training, note that in the proofs of Lemma 3.3 and Theorem 3.5 we show that gradient flow converges to a global minimizer of the loss. Let $\hat{\boldsymbol{y}} = \lim_{t\to\infty} \boldsymbol{X}\boldsymbol{W}_1 \boldsymbol{W}_2 \ldots \boldsymbol{W}_L$ denote the final network predictions. Then

$$\frac{\|\hat{\boldsymbol{y}}\|_2}{\|\boldsymbol{X}\|_{\text{op}}} \leq \lim_{t\to\infty} \|\boldsymbol{W}_1 \boldsymbol{W}_2 \ldots \boldsymbol{W}_L\|_2 \leq \lim_{t\to\infty} \prod_{l=1}^L s_l^2 \tag{72}$$

If $d \geq n$, $\hat{\boldsymbol{y}}$ is just equal to the vector of target outputs which is larger than zero by construction. If $d < n$, $\hat{\boldsymbol{y}}$ is the projection of the targets into the space spanned by the rows of $\boldsymbol{X}$, which is almost surely a non-zero vector. This implies that

$$\lim_{t\to\infty} \prod_{l=1}^L s_l^2 > 0 \tag{73}$$

which implies that the individual singular values are bounded away from zero at the end of training. In the population training case, the proof is nearly same, replacing $\hat{\boldsymbol{y}} = \lim_{t\to\infty} \boldsymbol{W}_1 \boldsymbol{W}_2 \ldots \boldsymbol{W}_L = \boldsymbol{\beta}_{\text{s}}$

$\square$

By Lemma D.1, we have

$$\boldsymbol{W}_1 \boldsymbol{W}_2 \ldots \boldsymbol{W}_{L-1} = c\boldsymbol{u}_1 \boldsymbol{v}_{L-1}^T \tag{74}$$

after pretraining, for some $c \in \mathbb{R}$. However, from Theorem 3.5 we know that after pretraining

$$\boldsymbol{W}_1 \ldots \boldsymbol{W}_{L-1} \boldsymbol{W}_L = \boldsymbol{\beta}_{\text{s}} \tag{75}$$

$$= c\boldsymbol{u}_1 (\boldsymbol{v}_{L_1}^T \boldsymbol{W}_L) \tag{76}$$

$$= c\boldsymbol{u}_1 \tag{77}$$

where we have used Lemma D.1 in the third equality to eliminate the inner product between the adjacent singular vectors. The possible factor of $-1$ can be absorbed into the definition of $\boldsymbol{u}_1$. This implies

$$\boldsymbol{W}_1 \boldsymbol{W}_2 \ldots \boldsymbol{W}_{L-1} = \boldsymbol{\beta}_{\text{s}} \boldsymbol{v}_{L-1}^T \tag{78}$$

### D.4. Proof of Theorem 3.5

This proof follows (Yun et al., 2021) closely but extends their result to the case $n > d$. We first show that gradient flow converges to a global minimum of the empirical loss (3). We then show that as $\alpha \to 0$, this minimum corresponds to the minimum norm least squares solution.

**Part 1**: Gradient flow converges to a global minimum

This proof follows the same logic as the proof for Lemma 3.3. First, we define the residual vector $\boldsymbol{r} = \boldsymbol{X}\boldsymbol{W}_1 \boldsymbol{W}_2 \ldots \boldsymbol{W}_L - \boldsymbol{y}_{\text{t}}$. Then we can write the empirical loss as

$$\mathcal{L} = \frac{1}{2n}\|\boldsymbol{r}\|_2^2 = \frac{1}{2n}(\|\boldsymbol{r}_\|\|_2^2 + \|\boldsymbol{r}_\perp\|_2^2) \tag{79}$$

where $r_\parallel$ is the component of $r$ in $\mathrm{im}(X)$ and $r_\perp$ is the component of $r$ in $\ker(X^T)$. Since $XW_1W_2\dots W_L \in \mathrm{im}(X)$, the global minimum of (79) is equal to $\|r_\perp\|_2^2$. Therefore, to show that gradient flow converges to a global minimum it is sufficient to show that $\lim_{t\to\infty}\|r_\parallel(t)\|_2^2 = 0$. Let $P_\parallel$ and $P_\perp$ be the orthogonal projectors onto $\mathrm{im}(X)$ and $\ker(X^T)$ respectively, so that $\mathcal{L}_\parallel := \|r_\parallel\|_2^2 = \|P_\parallel(XW_1W_2\dots W_L - y_t)\|_2^2$ and $\mathcal{L}_\perp := \|r_\perp\|_2^2 = \|P_\perp(XW_1W_2\dots W_L - y_t)\|_2^2$. Then we have

$$\dot{\mathcal{L}}_\parallel = \sum_{l=1}^{L}\langle \nabla_l \mathcal{L}_\parallel, \dot{W}_l\rangle \tag{80}$$

$$= -\sum_{l=1}^{L}\langle \nabla_l \mathcal{L}_\parallel, \nabla_l \mathcal{L}\rangle \tag{81}$$

$$= -\sum_{l=1}^{L}(\|\nabla_l \mathcal{L}_\parallel\|_F^2 + \langle \nabla_l \mathcal{L}_\parallel, \nabla_l \mathcal{L}_\perp\rangle) \tag{82}$$

Taking the gradient of $\mathcal{L}_\perp$ we have

$$\nabla_l \mathcal{L}_\perp = W_{l-1}^T \dots W_1^T X^T P_\perp r W_L^T \dots W_{l+1}^T = 0 \tag{83}$$

so

$$\dot{\mathcal{L}}_\parallel = -\sum_{l=1}^{L}\|\nabla_l \mathcal{L}_\parallel\|_F^2 \tag{84}$$

$$\leq -\|\nabla_L \mathcal{L}_\parallel\|_F^2 \tag{85}$$

$$= -\|W_{L-1}^T \dots W_1^T X^T P_\parallel r\|_2^2 \tag{86}$$

$$\leq -\sigma_{\min}^2(W_{L-1}^T \dots W_1^T)\|X^T P_\parallel r\|_2^2 \tag{87}$$

where $\sigma_{\min}(W_{L-1}^T \dots W_1^T)$ is the smallest singular value of $W_{L-1}^T \dots W_1^T$. From (43) - (47) we can bound this quantity away from zero. Then we have

$$\dot{\mathcal{L}}_\parallel \leq -(\alpha^2\lambda)^{L-1}\|X^T P_\parallel r\|_2^2 \tag{88}$$

$$\leq -2(\alpha^2\lambda)^{L-1}\lambda_{\min}\mathcal{L}_\parallel \tag{89}$$

where $\lambda_{\min}$ is the smallest nonzero eigenvalue of $XX^T$. The solution to the dynamics (89) is $\mathcal{L}_\parallel(t) \leq \mathcal{L}_\parallel(0)e^{-2(\alpha^2\lambda)^{L-1}\lambda_{\min}t}$, which proves $\lim_{t\to\infty}\|r_\parallel(t)\|_2^2 = 0$. Note that this part of the theorem holds for any $\alpha,n,d$, and we take the limit $\alpha \to 0$ after $t \to \infty$.

**Part 2**: as $\alpha \to 0$, gradient flow finds the minimum norm interpolator

In the case $n > d$, the least squares problem () is overdetermined so the solution is unique. That is, the unique solution is trivially the minimum norm solution. In the case $n \leq d$, there are multiple $\beta(t)$ that yield zero training error. Lemma D.1 shows that the parameter matrices are approximately rank one at all times and $u_{l+1}$ and $v_l$ align at the end of training as $\alpha \to 0$, which means that

$$\lim_{\alpha\to 0}\lim_{t\to\infty}\beta(t) = \lim_{\alpha\to 0}\lim_{t\to\infty}W_1W_2\dots W_L = cu_1 \tag{90}$$

where $c > 0$. Next we show that $u_l \in \mathrm{row}(X)$. We can break $W_1$ into two components $W_1^\parallel$ and $W_1^\perp$ where the columns of $W_1^\parallel$ are in $\mathrm{row}(X)$ and the columns of $W_1^\perp$ are in $\ker(X^T)$. The left hand side of (83) also shows that the gradient of $W_1^\perp$ is zero, which means that this component remains unchanged under gradient flow dynamics. Therefore we have

$$\|W_1^\perp(t)\|_F = \|W_1^\perp(0)\|_F \leq \alpha\|\bar{W}_1\|_F \tag{91}$$

which vanishes in the limit $\alpha \to 0$. This implies that $u_1 \in \mathrm{row}(X)$ at all times. The only global minimizer with this property is the minimum norm solution. As a final comment, we note that this theorem is also proven in Atanasov et al. (2021) using different techniques.

### D.5. Proof of Theorem 3.6

Let $\hat{\boldsymbol{\beta}} = \lim_{t \to \infty} \boldsymbol{W}_1 \boldsymbol{W}_2 \dots \boldsymbol{W}_L$. From Theorem 3.5, $\hat{\boldsymbol{\beta}} = \boldsymbol{X}^+ \boldsymbol{y} = \boldsymbol{X}^+ \boldsymbol{X} \boldsymbol{\beta}_\mathrm{t} + \boldsymbol{X}^+ \boldsymbol{\epsilon}$. Then the average generalization error at the end of training can be written

$$\mathbb{E}_{\boldsymbol{X}, \boldsymbol{\epsilon}} \mathcal{R} = \mathbb{E}_{\boldsymbol{X}, \boldsymbol{\epsilon}} \|\boldsymbol{\beta}_\mathrm{t} - \hat{\boldsymbol{\beta}}\|_2^2 \tag{92}$$

$$= 1 + \mathbb{E}_{\boldsymbol{X}, \boldsymbol{\epsilon}} \|\hat{\boldsymbol{\beta}}\|_2^2 - 2\mathbb{E}_{\boldsymbol{X}, \boldsymbol{\epsilon}} \langle \hat{\boldsymbol{\beta}}, \boldsymbol{\beta}_\mathrm{t} \rangle \tag{93}$$

$$= 1 + \mathbb{E}_{\boldsymbol{X}} \boldsymbol{\beta}_\mathrm{t}^T (\boldsymbol{X}^+ \boldsymbol{X})^T (\boldsymbol{X}^+ \boldsymbol{X}) \boldsymbol{\beta}_\mathrm{t} + \mathbb{E}_{\boldsymbol{X}, \boldsymbol{\epsilon}} \boldsymbol{\epsilon}^T (\boldsymbol{X}^+)^T \boldsymbol{X}^+ \boldsymbol{\epsilon} + 2\mathbb{E}_{\boldsymbol{X}, \boldsymbol{\epsilon}} \boldsymbol{\epsilon}^T (\boldsymbol{X}^+ \boldsymbol{X}) \boldsymbol{\beta}_\mathrm{t} \tag{94}$$

$$- 2(\mathbb{E}_{\boldsymbol{X}} \boldsymbol{\beta}_\mathrm{t}^T (\boldsymbol{X}^+ \boldsymbol{X}) \boldsymbol{\beta}_\mathrm{t} + \mathbb{E}_{\boldsymbol{X}, \boldsymbol{\epsilon}} \boldsymbol{\beta}_\mathrm{t}^T \boldsymbol{X}^+ \boldsymbol{\epsilon}) \tag{95}$$

$$= 1 - \mathbb{E}_{\boldsymbol{X}} \|\boldsymbol{P}_{\mathrm{row}(\boldsymbol{X})} \boldsymbol{\beta}_\mathrm{t}\|_2^2 + \sigma^2 \mathbb{E}_{\boldsymbol{X}} \mathrm{tr}((\boldsymbol{X}^+)^T \boldsymbol{X}^+) \tag{96}$$

where we have used the independence of $\boldsymbol{\epsilon}$ and $\boldsymbol{X}$, as well as the fact that the operator $\boldsymbol{X}^+ \boldsymbol{X}$ is the projector onto subspace spanned by the rows of $\boldsymbol{X}$, $\boldsymbol{P}_{\mathrm{row}(\boldsymbol{X})}$. Since the entries of the data matrix $\boldsymbol{X}$ are independent Gaussians, the n-dimensional subspace row$(\boldsymbol{X})$ is uniformly random in the Grassmanian manifold $\mathcal{G}_{n,d}$ (Vershynin, 2018), so $\boldsymbol{P}_{\mathrm{row}(\boldsymbol{X})} \boldsymbol{\beta}_\mathrm{t}$ is a random projection of $\boldsymbol{\beta}_\mathrm{t}$. Then

$$\mathbb{E}_{\boldsymbol{X}} \|\boldsymbol{P}_{\mathrm{row}(\boldsymbol{X})} \boldsymbol{\beta}_\mathrm{t}\|_2^2 = \gamma \tag{97}$$

which is a classic result in the theory of random projections (c.f. (Vershynin, 2018) Lemma 5.3.2). We now turn to the final term in (96). Let $\{\sigma_l\}_{l \le \min(n,d)}$ be the nonzero singular values of the data matrix $\boldsymbol{X}$. Then

$$\mathbb{E}_{\boldsymbol{X}} \mathrm{tr}((\boldsymbol{X}^+)^T \boldsymbol{X}^+) = \mathbb{E}_{\boldsymbol{X}} \sum_{l=1}^{\min(n,d)} \frac{1}{\sigma_l^2} \tag{98}$$

First take the case $\gamma < 1$. Then there are $n$ nonzero singular values of $\boldsymbol{X}$, which are the eigenvalues of the Wishart matrix $\boldsymbol{C} = \frac{1}{d} \boldsymbol{X} \boldsymbol{X}^T$ and

$$\mathbb{E}_{\boldsymbol{X}} \mathrm{tr}((\boldsymbol{X}^+)^T \boldsymbol{X}^+) = \frac{\gamma}{n} \mathbb{E}_{\boldsymbol{X}} \mathrm{tr}(\boldsymbol{C}^{-1}) \tag{99}$$

$$= -\gamma \lim_{z \to 0} \frac{1}{n} \mathbb{E}[\mathrm{tr}((z\boldsymbol{I} - \boldsymbol{C})^{-1})] \tag{100}$$

$$= -\gamma \lim_{z \to 0} \mathfrak{g}_{\boldsymbol{C}}(z) \tag{101}$$

In the second line we have introduced the complex variable $z$, which casts the quantity of interest as the $z \to 0$ limit of the normalized expected trace of the resolvent of $\boldsymbol{C}$. In the limit of large $n$, this quantity tends to the Stieltjes transform of the Wishart matrix $\mathfrak{g}_{\boldsymbol{C}}(z)$, which has a closed form expression (see (Potters & Bouchaud, 2020) Ch.4 for a proof).

$$\lim_{z \to 0} \mathfrak{g}_{\boldsymbol{C}}(z) = \lim_{z \to 0} \frac{z - (1 - \gamma) - \sqrt{z - (1 + \sqrt{\gamma})^2} \sqrt{z - (1 - \sqrt{\gamma})^2}}{2\gamma z} \tag{102}$$

$$= -\frac{1}{1 - \gamma} \tag{103}$$

so $\mathbb{E}_{\boldsymbol{X}} \mathrm{tr}((\boldsymbol{X}^+)^T \boldsymbol{X}^+) = \frac{\gamma}{1-\gamma}$ for $\gamma < 1$. In the case $\gamma > 1$, there will be $d$ terms in the sum (98), which are proportional to the eigenvalues of the covariance matrix $\frac{1}{n} \boldsymbol{X}^T \boldsymbol{X}$. If we define $n' = d, d' = n, \gamma' = n'/d'$ and $\boldsymbol{X}' = \boldsymbol{X}^T \in \mathbb{R}^{n' \times d'}$, equations (99) - (101) hold under the substitution $\gamma \to \gamma'$. So $\mathbb{E}_{\boldsymbol{X}} \mathrm{tr}((\boldsymbol{X}^+)^T \boldsymbol{X}^+) = \frac{\gamma'}{1-\gamma'} = \frac{1}{\gamma-1}$ for $\gamma > 1$. Putting everything together we have

$$\mathbb{E}_{\boldsymbol{X}, \boldsymbol{\epsilon}} \mathcal{R} = \begin{cases} \frac{(1-\gamma)^2 + \gamma \sigma^2}{1-\gamma} & \gamma < 1 \\ \frac{\sigma^2}{\gamma - 1} & \gamma > 1 \end{cases} \tag{104}$$

### D.6. Proof of Theorem 3.7

Theorem 3.4 implies that the pretrained feature matrix is $\boldsymbol{\Phi} = (\boldsymbol{X} \boldsymbol{\beta}_\mathrm{s}) \boldsymbol{v}_{L-1}^T$. Since $\boldsymbol{\Phi}$ is a rank one matrix its pseudoinverse is easy to compute

$$\boldsymbol{\Phi}^+ = \frac{1}{\|\boldsymbol{X} \boldsymbol{\beta}_\mathrm{s}\|_2^2} \boldsymbol{v}_{L-1} (\boldsymbol{X} \boldsymbol{\beta}_\mathrm{s})^T \tag{105}$$

The coefficent vector $\hat{\beta}$ after linear transfer is

$$\hat{\beta} = W_1 \dots W_{L-1} \hat{W}_L \tag{106}$$

$$= W_1 \dots W_{L-1} \Phi^+ y_t \tag{107}$$

$$= b \beta_s \tag{108}$$

where

$$b = \frac{\beta_s^T X^T y_t}{\beta_s^T X^T X \beta_s} \tag{109}$$

$$= \frac{\beta_s^T X^T X \beta_t}{\beta_s^T X^T X \beta_s} + \frac{\beta_s^T X^T \epsilon}{\beta_s^T X^T X \beta_s} \tag{110}$$

As in the proof of Theorem 3.6, we can write the typical generalization error as

$$\mathbb{E}_{X,\epsilon} \mathcal{R}_{lt} = \|\hat{\beta} - \beta_t\|_2^2 \tag{111}$$

$$= 1 + \mathbb{E}_{X,\epsilon} b^2 - 2\cos\theta \mathbb{E}_{X,\epsilon} b \tag{112}$$

To proceed, we can write $\beta_t = \cos\theta \beta_s + \sin\theta \nu$ for some vector $\nu \perp \beta_s$, and introduce the independent $n-$dimensional Gaussian vectors $z = X\beta_s \sim \mathcal{N}(0, I_n)$ and $w = X\nu \sim \mathcal{N}(0, I_n)$. With this change of variables we have

$$\mathbb{E}_{X,\epsilon} b = \mathbb{E}_{z,w,\epsilon} b \tag{113}$$

$$= \cos\theta \tag{114}$$

$$\mathbb{E}_{X,\epsilon} b^2 = \mathbb{E}_{z,w,\epsilon} b^2 \tag{115}$$

$$= \cos^2\theta + (\sin^2\theta + \sigma^2)\mathbb{E}_z \frac{1}{\|z\|_2^2} \tag{116}$$

The integral $\mathbb{E}_z \frac{1}{\|z\|_2^2}$ can be solved exactly

$$\mathbb{E}_z \frac{1}{\|z\|_2^2} = \frac{1}{(2\pi)^{n/2}} \int_{-\infty}^{\infty} \frac{e^{-\sum_{i=1}^n z_i^2/2}}{\sum_{j=1}^n z_j^2} dz \tag{117}$$

$$= \frac{S_{n-1}}{(2\pi)^{n/2}} \int_0^{\infty} r^{n-3} e^{r^2/2} dr \tag{118}$$

$$= \frac{S_{n-1}}{4\pi^{n/2}} \int_0^{\infty} e^{-t} t^{\frac{n}{2}-2} dt \tag{119}$$

$$= \frac{S_{n-1}}{4\pi^{n/2}} \Gamma\left(\frac{n}{2} - 1\right) \tag{120}$$

$$= \frac{1}{n-2} \tag{121}$$

which completes the proof.

### D.7. Proof of Theorem 3.8

We begin by writing down the solution to the optimization problem (12)

$$\hat{W}_L = (\Phi^T \Phi + n\lambda I_d)^{-1} \Phi^T y_t \tag{122}$$

As in the proof of Theorem 3.7, we have

$$\Phi = (X\beta_s) v_{L-1}^T \tag{123}$$

$$W_1 W_2 \dots W_{L-1} = \beta_s v_{L-1}^T \tag{124}$$

Combining these expressions we can solve for the linear function the network implements after transfer learning with ridge regression

$$\hat{\boldsymbol{\beta}} = \boldsymbol{W}_1 \boldsymbol{W}_2 \ldots \boldsymbol{W}_{L-1} \hat{\boldsymbol{W}}_L \tag{125}$$

$$= \boldsymbol{\beta}_s \boldsymbol{v}_{L-1}^T (\|\boldsymbol{X}\boldsymbol{\beta}_s\|_2^2 + n\lambda \boldsymbol{I}_d)^{-1} \boldsymbol{v}_{L-1} (\boldsymbol{X}\boldsymbol{\beta}_s)^T \boldsymbol{y}_t \tag{126}$$

$$= \left( \frac{(\boldsymbol{X}\boldsymbol{\beta}_s)^T \boldsymbol{y}_t}{\|\boldsymbol{X}\boldsymbol{\beta}_s\|_2^2 + n\lambda} \right) \boldsymbol{\beta}_s \tag{127}$$

As in the proof of Theorem 3.7, we write $\boldsymbol{\beta}_t = \cos\theta \boldsymbol{\beta}_s + \sin\theta \boldsymbol{\nu}$ for some vector $\boldsymbol{\nu} \perp \boldsymbol{\beta}_s$, and introduce the independent $n-$dimensional Gaussian vectors $\boldsymbol{z} = \boldsymbol{X}\boldsymbol{\beta}_s \sim \mathcal{N}(0, \boldsymbol{I}_n)$ and $\boldsymbol{w} = \boldsymbol{X}\boldsymbol{\nu} \sim \mathcal{N}(0, \boldsymbol{I}_n)$. Then we can get the following expression for the generalization error of ridge linear transfer:

$$\mathbb{E}_{\boldsymbol{X},\epsilon} \mathcal{R}_{lt}^{\lambda} = \|\hat{\boldsymbol{\beta}} - \boldsymbol{\beta}_t\|_2^2 \tag{128}$$

$$= 1 + (\cos^2\theta) I_1(n+2, \lambda) + (\sin^2\theta + \sigma^2) I_1(n, \lambda) - (2\cos^2\theta) I_2(n, \lambda) \tag{129}$$

where we have used spherical coordinates to define the following integrals

$$I_1(m, \lambda) = \mathbb{E}_z \left( \frac{\|z\|_2^{m-n+2}}{(\|z\|_2^2 + n\lambda)^2} \right) = \frac{S_{n-1}}{(2\pi)^{n/2}} \int_0^{\infty} \frac{r^{m+1} e^{-r^2/2}}{(r^2 + n\lambda)^2} dr \tag{130}$$

$$I_2(m, \lambda) = \mathbb{E}_z \left( \frac{\|z\|_2^{m-n+2}}{\|z\|_2^2 + n\lambda} \right) = \frac{S_{n-1}}{(2\pi)^{n/2}} \int_0^{\infty} \frac{r^{m+1} e^{-r^2/2}}{r^2 + n\lambda} dr \tag{131}$$

We evaluate $I_1(n, \lambda), I_1(n+2, \lambda)$ and $I_2(n, \lambda)$ for large $n$. To avoid cluttering the notation, we ignore the coefficient $\frac{S_{n-1}}{(2\pi)^{n/2}}$ while solving the integral and restore it at the end of the calculation. Then

$$I_1(n, \lambda) \propto 2^{n/2} \int \frac{u^{n/2} e^{-u}}{(2u + n\lambda)^2} du \tag{132}$$

$$= n(2n)^{n/2} \int \frac{t^{n/2} e^{-nt}}{(2nt + n\lambda)^2} dt \tag{133}$$

$$= n(2n)^{n/2} \int g(t) e^{nf(t)} dt \tag{134}$$

$$\approx n(2n)^{n/2} \sqrt{\frac{2\pi}{n|f''(t_0)|}} g(t_0) e^{nf(t_0)} \tag{135}$$

We have introduced the change of variables $u = r^2/2$ in the first line, $t = u/n$ in the second line, and finally evaluated the integral for large $n$ using the saddle point method. In the last line, $t_0$ is a critical point of $f(t) = \frac{1}{2} \log t - t$ and $g(t) = (2nt + n\lambda)^{-2}$. Differentiating $f(t)$ and setting equal to zero we find $t_0 = 1/2$. So for large $n$,

$$I_1(n, \lambda) \propto \frac{\sqrt{\pi n} n^{n/2} e^{-n/2}}{(n + n\lambda)^2} \tag{136}$$

We can now restore the angular coefficient to the integral

$$I_1(n, \lambda) = \frac{S_{n-1}}{(2\pi)^{n/2}} \frac{\sqrt{\pi n} n^{n/2} e^{-n/2}}{(n + n\lambda)^2} \tag{137}$$

$$\approx \frac{n\pi^{n/2}}{\sqrt{\pi n}} \left( \frac{n}{2} \right)^{-n/2} e^{n/2} \frac{\sqrt{\pi n} n^{n/2} e^{-n/2}}{(n + n\lambda)^2} \tag{138}$$

$$= \frac{1}{n(1 + \lambda)^2} \tag{139}$$

where we have used Stirling's approximation in the second line. Therefore, $\lim_{n\to\infty} I_1(n, \lambda) = 0$. We stress that although the integral was approximated at the saddle point, the limit $n \to \infty$ is exact since corrections to the saddle point value are

subleading in $n$. Similar calculations yield

$$I_1(n+2, \lambda) = \frac{1}{(1+\lambda)^2} \tag{140}$$

$$I_2(n, \lambda) = \frac{1}{1+\lambda} \tag{141}$$

for large $n$. Plugging this into (128), we have

$$\lim_{n\to\infty} \mathbb{E}_{\boldsymbol{X},\boldsymbol{\epsilon}} \mathcal{R}_{lt}^{\lambda} = 1 - \frac{(1+2\lambda)}{(1+\lambda)^2} \cos^2\theta \tag{142}$$

This is a strictly increasing function in $\lambda \geq 0$ for any $\theta \in [0, \pi/2]$, which implies that the optimal regularization value is $\lambda^* = 0$.

### D.8. Proof of Theorem 3.9

The proof involves slightly tweaking the proof of Theorem 3.5. Since the source trained model obeyed the initialization assumption (19), the invariant matrix (36) is equal to its value at initialization before pretraining throughout fine tuning as well. This implies that the first half of the proof of Theorem (3.5) holds in the fine tuning case and the model will converge to a global minimizer of the training loss. The invariance throughout fine tuning also implies that (90) holds and that $\boldsymbol{W}_1^{\perp}$ does not change during fine tuning, and remains fixed at its initial value from pretraining. Therefore, by the proof of Theorem 3.7, at the beginning of fine tuning, $\boldsymbol{u}_1 = \boldsymbol{\beta}_s$ and $(\boldsymbol{I} - \boldsymbol{P}_{\mathrm{row}(\boldsymbol{X})})\boldsymbol{\beta}_s$ is the component of $\boldsymbol{u}_1$ that does not evolve. Meanwhile, $\boldsymbol{P}_{\mathrm{row}}(\boldsymbol{X})\boldsymbol{u}_1$ will evolve to the minimum norm solution. Combining these results, after fine tuning,

$$\lim_{\alpha\to 0} \lim_{t\to\infty} \boldsymbol{\beta}_{ft}(t) = \boldsymbol{\beta}_{sc} + (\boldsymbol{I} - \boldsymbol{P}_{\mathrm{row}(\boldsymbol{X})})\boldsymbol{\beta}_s \tag{143}$$

where $\boldsymbol{\beta}_{sc}$ is the minimum norm solution. We can now write the expected generalization error

$$
\begin{aligned}
\mathbb{E}_{\boldsymbol{X},\boldsymbol{\epsilon}}\mathcal{R}_{ft} &= \mathbb{E}_{\boldsymbol{X},\boldsymbol{\epsilon}}[\|\boldsymbol{\beta}_{\mathrm{t}} - \boldsymbol{\beta}_{\mathrm{ft}}\|_2^2] \\
&= \mathbb{E}_{\boldsymbol{X},\boldsymbol{\epsilon}}\mathcal{R}_{sc} + \mathbb{E}_{\boldsymbol{X}}\|(\boldsymbol{I} - \boldsymbol{P}_{\mathrm{row}(\boldsymbol{X})})\boldsymbol{\beta}_s\|_2^2 - 2\mathbb{E}_{\boldsymbol{X}}\langle\boldsymbol{\beta}_{\mathrm{t}}, (\boldsymbol{I} - \boldsymbol{P}_{\mathrm{row}(\boldsymbol{X})})\boldsymbol{\beta}_s\rangle \\
&= \mathbb{E}_{\boldsymbol{X},\boldsymbol{\epsilon}}\mathcal{R}_{sc} + \max(0, 1-\gamma) - 2\mathbb{E}_{\boldsymbol{X}}\langle\boldsymbol{\beta}_{\mathrm{t}}, (\boldsymbol{I} - \boldsymbol{P}_{\mathrm{row}(\boldsymbol{X})})\boldsymbol{\beta}_s\rangle \\
&= \mathbb{E}_{\boldsymbol{X},\boldsymbol{\epsilon}}\mathcal{R}_{sc} + \max(0, 1-\gamma) - 2\cos\theta\mathbb{E}_{\boldsymbol{X}}\langle\boldsymbol{\beta}_s, (\boldsymbol{I} - \boldsymbol{P}_{\mathrm{row}(\boldsymbol{X})})\boldsymbol{\beta}_s\rangle \\
&\quad - 2\sin\theta\mathbb{E}_{\boldsymbol{X}}\langle\boldsymbol{\nu}, (\boldsymbol{I} - \boldsymbol{P}_{\mathrm{row}(\boldsymbol{X})})\boldsymbol{\beta}_s\rangle \\
&= \mathbb{E}_{\boldsymbol{X},\boldsymbol{\epsilon}}\mathcal{R}_{sc} + \max(0, 1-\gamma)(1 - 2\cos\theta) - 2\mathbb{E}_{\boldsymbol{X}}\sin\theta\langle\boldsymbol{\nu}, (\boldsymbol{I} - \boldsymbol{P}_{\mathrm{row}(\boldsymbol{X})})\boldsymbol{\beta}_s\rangle
\end{aligned}
$$

where we have used the fact that $\boldsymbol{P}_{\mathrm{row}(\boldsymbol{X})})\boldsymbol{\beta}_s$ is a random projection as in the proof of Theorem 3.6 and set $\boldsymbol{\beta}_{\mathrm{t}} = \cos\theta\boldsymbol{\beta}_s + \sin\theta\boldsymbol{\nu}$ for some $\boldsymbol{\nu} \perp \boldsymbol{\beta}_s$. The final term is equal to zero for the following reason. The operator $\boldsymbol{I} - \boldsymbol{P}_{\mathrm{row}(\boldsymbol{X})}$ is a random projector onto the $d - n$ dimensional subspace orthogonal to row$\boldsymbol{X}$ Since the uniform distribution of random subspaces is rotationally invariant, we can instead fix a particular subspace and average over $\boldsymbol{\beta}_s \sim \mathrm{Uniform}(S^{d-1})$. Using rotation invariance again, we can fix the projection to be along the first $d - n$ coordinates of $\boldsymbol{\beta}_s$. Then we have

$$\mathbb{E}\langle\boldsymbol{\nu}, (\boldsymbol{I} - \boldsymbol{P}_{\mathrm{row}(\boldsymbol{X})})\boldsymbol{\beta}_s\rangle = \sum_{k=1}^{n-d} \boldsymbol{\nu}_k \mathbb{E}(\boldsymbol{\beta}_s)_k \tag{144}$$

$$= 0 \tag{145}$$

This completes the proof

### D.9. Proof of Theorem C.1

To begin note that by Theorem 3.6, the network will implement the least squares solution to the source task when it is trained with gradient flow on a finite size data set. Denote this vector $\hat{\boldsymbol{\beta}}_{\mathrm{bm}} = \boldsymbol{X}_s^+ \boldsymbol{y}_s$. By the same reasoning as in Appendix D.8, after fine tuning

$$\lim_{\alpha\to 0} \lim_{t\to\infty} \boldsymbol{\beta}_{\mathrm{ft}}(t) = \boldsymbol{\beta}_{\mathrm{sc}} + (\boldsymbol{I} - \boldsymbol{P}_{\mathrm{t}})\hat{\boldsymbol{\beta}}_s \tag{146}$$

where $P_t$ is the orthogonal projector onto the space spanned by the rows of $X_t$. With this expression we can write down the expected generalization error of the fine-tuned model

$$\mathbb{E}_{X_s,X_t,\epsilon_s,\epsilon_t} \mathcal{R}_{ft} = \mathbb{E}_{X_s,X_t,\epsilon_s,\epsilon_t} \|\beta_t - \beta_{sc} - (I - P_t)\hat{\beta}_s\|_2^2 \tag{147}$$

$$= \mathbb{E}_{X_t,\epsilon_t} \|\beta_t - \beta_{sc}\|_2^2 + \mathbb{E}_{X_s,X_t,\epsilon_s} \|(I - P_t)\hat{\beta}_s\|_2^2 - 2\mathbb{E}_{X_s,X_t,\epsilon_s}\beta_t^T(I - P_t)\hat{\beta}_s \tag{148}$$

$$= \mathbb{E}_{\mathcal{D}_t}\mathcal{R}_{sc} + \max(0, 1 - \gamma_t)\mathbb{E}_{X_s\epsilon_s} \|\hat{\beta}_s\|_2^2 - 2\mathbb{E}_{X_s,X_t,\epsilon_s}\beta_t^T(I - P_t)\hat{\beta}_s \tag{149}$$

where we have used the result on random projections employed in the proof of Theorem 3.6 along with the independence of the matrices $X_s$ and $X_t$. Let $P_s$ denote the orthogonal projector onto the rowspace of $X_s$. Then we can write

$$\mathbb{E}_{X_s,\epsilon_s} \|\hat{\beta}_s\|_2^2 = \mathbb{E}_{X_s,\epsilon_s} \|P_s\beta_s + X_s^+\epsilon_s\|_2^2 \tag{150}$$

$$= \mathbb{E}_{X_s,\epsilon_s} \|P_s\beta_s\|_2^2 + \sigma_s^2\mathbb{E}_{X_s}\text{tr}(X_s^{+T}X_s^+) \tag{151}$$

$$= \begin{cases} \gamma_s + \frac{\sigma_s^2\gamma_s}{1-\gamma_s} & \gamma_s < 1 \\ 1 + \frac{\sigma_s^2}{\gamma_s-1} & \gamma_s > 1 \end{cases} \tag{152}$$

where the final line follows from the same reasoning as in the proof of Theorem 3.6 (Appendix D.5). The final term in 149 can be computed using similar techniques to those in Appendix D.8. Let $\beta_t = a\nu + b\hat{\beta}_s$ for some $\nu \perp \hat{\beta}_s$ in the subspace spanned by $\beta_t$ and $\hat{\beta}_s$. Then

$$\mathbb{E}_{X_s,X_t,\epsilon_s}\beta_t^T(I - P_t)\hat{\beta}_s = \mathbb{E}_{X_s,\epsilon_2}a\mathbb{E}_{X_t}\nu^T(I - P_t)\hat{\beta}_s + \mathbb{E}_{X_s,\epsilon_2}b\mathbb{E}_{X_t}\hat{\beta}_s^T(I - P_t)\hat{\beta}_s \tag{153}$$

$$= \mathbb{E}_{X_s,\epsilon_2}b\mathbb{E}_{X_t}\hat{\beta}_s^T(I - P_t)\hat{\beta}_s \tag{154}$$

$$= \mathbb{E}_{X_s,\epsilon_2}b\mathbb{E}_{X_t} \|(I - P_t)\hat{\beta}_s\|_2^2 \tag{155}$$

$$= \max(0, 1 - \gamma_t)\mathbb{E}_{X_s,\epsilon_2}b\|\hat{\beta}_s\|_2^2 \tag{156}$$

$$= \max(0, 1 - \gamma_t)\mathbb{E}_{X_s}\beta_t P_s\beta_s \tag{157}$$

$$= \cos\theta \max(0, 1 - \gamma_t)\min(1, \gamma_s) \tag{158}$$

In the second line we have used the same reasoning as at the end of Appendix D.8, in the third line we have used the properties of orthogonal projectors, and in the remaining lines we have used properties of random projections that are discussed in detail in Appendix D.5. Putting the results above together completes the proof.

## E. ReLU networks

In this section, we describe how to compute projections into (and out of) the RKHS defined by a one hidden layer ReLU network. Consider a network $f(x)$ and a target function $f_*(x)$.

$$f(x) = \frac{1}{m}\sum_{i=1}^{m} c_i\sigma(w_i^T x) \tag{159}$$

$$f_*(x) = \frac{1}{m_*}\sum_{i=1}^{m_*} c_i^*\sigma(w_i^{*T} x) \tag{160}$$

The feature space of the model is $\text{span}\{\sigma(w_i^T x)\}_{i\leq m}$ in $L_2(p)$. To form projectors into this space and its orthogonal complement, we introduce the Mercer decomposition. For any positive definite, symmetric kernel $k : \mathcal{X} \times \mathcal{X} \to \mathbb{R}$ we can define features through partial evaluation of the kernel, i.e., $\phi(x) = k(\cdot, x)$. This kernel also induces a reproducing kernel Hilbert space (RKHS) via the Moore–Aronszajn theorem, which is defined as the set of all functions that are linear combinations of these features,

$$\mathcal{H}_k = \left\{ f \,\Big|\, f = \sum_{i=1}^{M} \alpha_i k(\cdot, z_i) \text{ for some } M \in \mathbb{N}, \alpha_i \in \mathbb{R}, z_i \in \mathcal{X} \right\} \tag{161}$$

The associated norm of a function $f \in \mathcal{H}_k$ is given by

$$||f||_k^2 = \sum_{ij}^{M} \alpha_i k(\boldsymbol{z}_i, \boldsymbol{z}_j) \alpha_j \tag{162}$$

We can also define the operator $T_k : L_2(p) \to L_2(p)$ with action

$$T_k f = \int d\boldsymbol{x}' p(\boldsymbol{x}') k(\boldsymbol{x}, \boldsymbol{x}') f(\boldsymbol{x}') \tag{163}$$

The spectral decomposition of this operator, $\{\lambda_l^2, \psi_l\}_{l=1}^{\infty}$ is known as the Mercer decomposition and the eigenfunctions form a basis for $L_2(p)$. The eigenfunctions $\psi_l(\boldsymbol{x})$ satisfy

$$T_k \psi_l = \lambda_l \psi_l \tag{164}$$

where $\lambda_l$ is the associated eigenvalue. The eigenfunctions with non-zero eigenvalue form a basis for the RKHS $\mathcal{H}_k$. Given a function $f = \sum_{l=1}^{\infty} c_l \psi_l$ one can show by direct computation that

$$||f||_k^2 = \sum_{l=1}^{\infty} c_l^2 / \lambda_l^2 \tag{165}$$

which also demonstrates that functions with support on eigenmodes with zero eigenvalue are not in the RKHS. If we can construct the Mercer eigenfunctions we can build orthogonal projection operators into the RKHS and its orthogonal complement. To begin note that for Gaussian data, $p(\boldsymbol{x}) = \mathcal{N}(0, \boldsymbol{I}_d)$, we can exactly compute the expected overlap between two ReLU functions in terms of their weight vectors (Cho & Saul, 2009):

$$\langle \sigma(\boldsymbol{w}_i^T \boldsymbol{x}) \sigma(\boldsymbol{w}_j^T \boldsymbol{x}) \rangle_{L_2} = \int p(\boldsymbol{x}) \sigma(\boldsymbol{w}_i^T \boldsymbol{x}) \sigma(\boldsymbol{w}_j^T \boldsymbol{x}) \tag{166}$$

$$= \frac{1}{2\pi} \left( \sqrt{1 - u_{ij}^2} + u(\pi - \arccos u_{ij}) \right) \tag{167}$$

where $u_{ij} = \frac{\boldsymbol{w}_i^T \boldsymbol{w}_j}{\|\boldsymbol{w}_i\|_2 \|\boldsymbol{w}_j\|_2}$ With this in hand, we can define the following matrices:

$$\boldsymbol{K}_{ij} = \frac{1}{m} \langle \sigma(\boldsymbol{w}_i^T \boldsymbol{x}) \sigma(\boldsymbol{w}_j^T \boldsymbol{x}) \rangle_{L_2} \tag{168}$$

$$\boldsymbol{K}_{ij}^* = \frac{1}{m_*} \langle \sigma(\boldsymbol{w}_i^{*T} \boldsymbol{x}) \sigma(\boldsymbol{w}_j^{*T} \boldsymbol{x}) \rangle_{L_2} \tag{169}$$

$$\tilde{\boldsymbol{K}}_{ij} = \frac{1}{\sqrt{mm_*}} \langle \sigma(\boldsymbol{w}_i^T \boldsymbol{x}) \sigma(\boldsymbol{w}_j^{*T} \boldsymbol{x}) \rangle_{L_2} \tag{170}$$

The Mercer eigenfunctions can be constructed by diagonalizing the matrix $K$. If $\boldsymbol{z}_l$ is an eigenvector of $K$ with eigenvalue $\lambda_l^2$, then

$$\psi_l(\boldsymbol{x}) = \frac{1}{\sqrt{m\lambda_l^2}} \sum_{l=1}^{m} (z_l)_i \sigma(\boldsymbol{w}_i^T \boldsymbol{x}) \tag{171}$$

is a Mercer eigenfuction with eigenvalue $\lambda_l^2$, which can be verified by plugging the expression into the eigenvalue equation (164). Since the feature space is $m$-dimensional, we know that these $m$ eigenfunctions span the RKHS. We can now write down expressions for the projections of $f_*(\boldsymbol{x})$ into this space and its orthogonal complement

$$||P_{\parallel} f_*(\boldsymbol{x})||_{L_2}^2 = \frac{1}{m_*} \boldsymbol{c}_*^T \tilde{\boldsymbol{K}}^T \boldsymbol{K}^{-1} \tilde{\boldsymbol{K}} \boldsymbol{c}_* \tag{172}$$

$$||P_{\perp} f_*(\boldsymbol{x})||_{L_2}^2 = ||f_*||_{L_2}^2 - ||P_{\parallel} f_*(\boldsymbol{x})||_{L_2}^2 = \frac{1}{m_*} \boldsymbol{c}_*^T \boldsymbol{K}_* \boldsymbol{c}_* - \frac{1}{m_*} \boldsymbol{c}_*^T \tilde{\boldsymbol{K}}^T \boldsymbol{K}^{-1} \tilde{\boldsymbol{K}} \boldsymbol{c}_* \tag{173}$$

## F. Experimental details

### F.1. Deep linear models

For the experiments in deep linear models, we train a two layer linear network with dimension $d = 500$. We initialize the weight matrices with random normal weights and scale parameter $\alpha = 10^{-5}$. To approximate gradient flow, we use full batch gradient descent with small learning rate $\eta = 10^{-3}$. We train each model for $10^5$ steps or until the training loss reaches $10^{-6}$. We perform target training for 20 instances of the training data and a grid of dataset sizes and values of $\theta$

### F.2. ReLU networks

For the experiments in shallow ReLU networks, we use the parameters $d = 100$, $m = 1000$, $m_* = 100$. We initialize the weight matrices randomly on the sphere and the output weights are initialized at $10^{-7}$. We approximate gradient flow with full batch gradient descent and learning rate $0.01m$ and train for $10^5$ iterations or until the loss reaches $10^{-6}$. For training with a finite dataset we use 100 realizations of the training data, and average over 10 random initialization seeds.

## G. Additional Figures

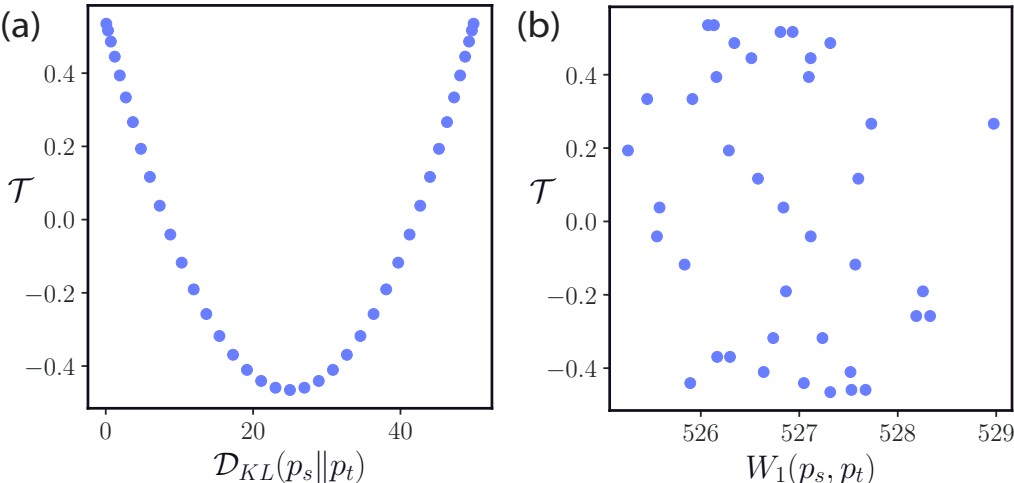

*Figure 4.* **Transferability is not predicted by $\phi$-divergences or integral probability metrics** We generate source and target distributions $p_s$, $p_t$ according to the setup in Section 3 and plot the transferability $\mathcal{T}$ (5) as a function of **(a)** the KL divergence $D_{\mathrm{KL}}(p_s \| p_t)$ and **(b)** the Wasserstein 1-metric. The KL divergence can be computed exactly in this setting (see Section D.1). $W_1$ is computed from finite samples using the algorithm in Sriperumbudur et al. (2009).

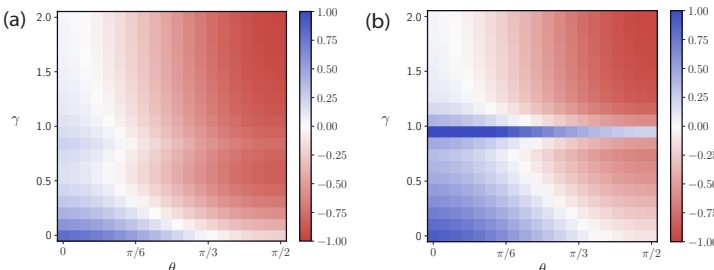

*Figure 5.* **Regularizing scratch training eliminates anomalous positive transfer**. Simulated linear transfer phase diagram for $L = 2$, $\sigma = 0.2$, $d = 500$ **(a)** with optimal weight decay in the scratch training and **(b)** without. To tune the weight decay hyperparameter, we sweep over a grid of $\lambda_{\text{wd}} \in [0, 10^{-4}, 10^{-3}, 10^{-2}, 10^{-1}]$ and choose the model that has the lowest generalization error. The transfer learning procedure is identical to Fig. 1, only scratch training is altered. In the regularized plot **(a)**, the spike of positive transfer along $\gamma = 1$ is eliminated, as the regularized scratch trained model does not undergo double descent.

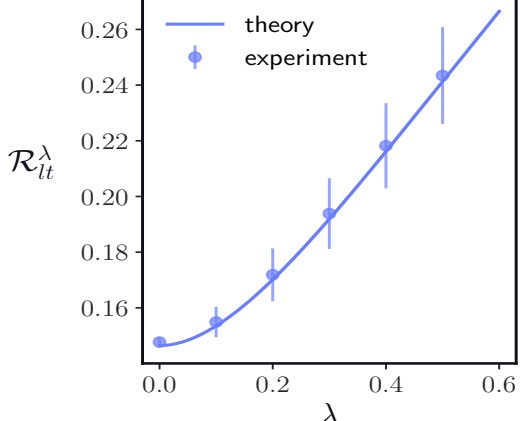

*Figure 6.* **Ridge regularization leads to worse generalization in linear transfer**. Linear transfer generalization error for $\gamma = 0.5$ as a function of regularization parameter $\lambda$. The generalization error is a strictly increasing function of $\lambda$, which implies that the optimal regularizer is $\lambda_* = 0$. Solid line is theory (3.8), points are experiments. Error bars represent the standard deviation over 20 realizations of the target dataset.

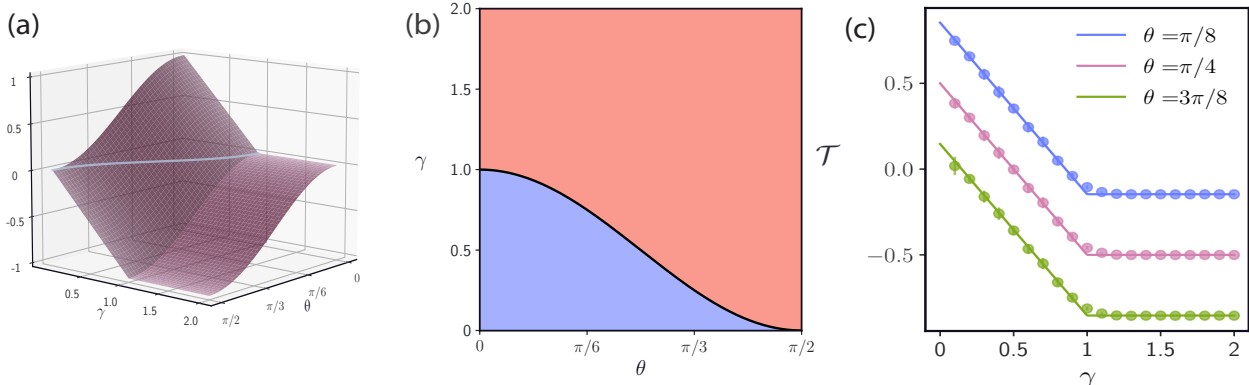

*Figure 7.* **Linear transferability, $\sigma = 0$** We pretrain a linear network (7) with $L = 2$ and $d = 500$ to produce un-noised labels from linear source function $y = \boldsymbol{\beta}_s^T \boldsymbol{x}$ using the population loss (2). We then retrain the final layer weights on a sample of $n = \gamma d$ points $(\boldsymbol{x}_i, y_i = \boldsymbol{\beta}_t^T \boldsymbol{x}_i)$ where $\boldsymbol{\beta}_s^T \boldsymbol{\beta}_t = \cos \theta$ and compare its generalization error to that of a model trained from scratch on the target dataset. **(a)** Theoretical transferability surface (5) as a function of the number of data points $\gamma = n/d$ and task overlap $\theta$. **(b)** Top-down view of (a), shaded by sign of transferability. Red indicates negative transferability $\mathcal{T} < 0$ and blue indicates positive transferability $\mathcal{T} > 0$. Note that transfer is always negative when $\gamma > 1$, since the scratch trained model can perfectly learn the target task as there is no label noise. **(c)** Slices of (a) for constant $\theta$. Solid lines are theory, dots are from numerical experiments. Error bars represent the standard deviation over 20 draws of the training data.

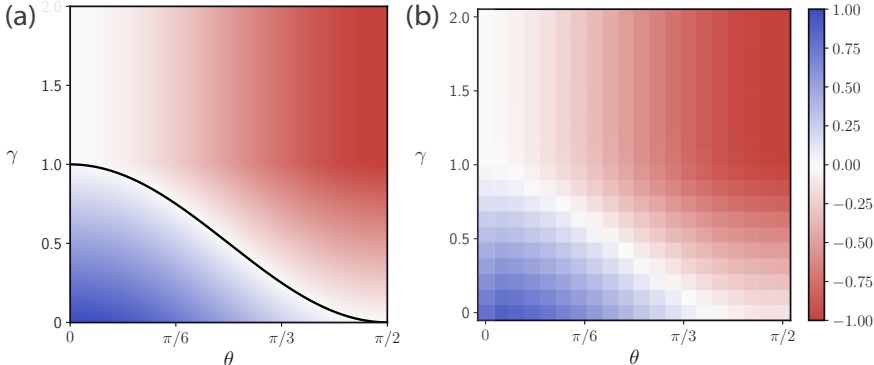

*Figure 8.* **Linear transfer $\sigma = 0$: theory vs. experiment (a)** Identical to Fig. 7(b), but shaded according to the value of the transferability. **(b)** Results of numerical simulations with $L = 2$, $d = 500$

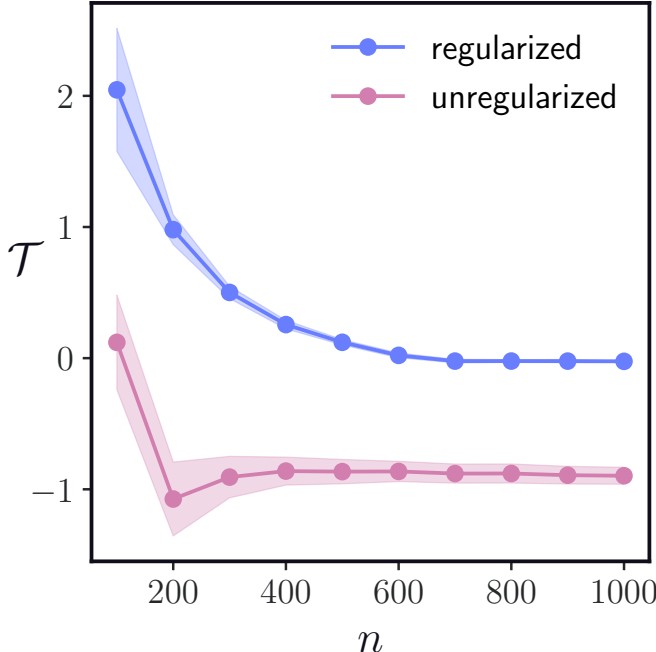

*Figure 9.* **Regularizing pretrained models toward the lazy regime eliminates negative transfer**: We train a two layer ReLU network on the transfer learning task defined by (16) and (17) with $\mu = 0.9$, $m = 1000$, $m_* = 100$, $d = 100$. During pretraining, we include a regularization term $\lambda \sum_{i=1}^{m} \|\boldsymbol{w}_i - \boldsymbol{w}_i^{(0)}\|_2^2$ where $\boldsymbol{w}_i^{(0)}$ is the random initial value of weight vector $\boldsymbol{w}_i$. This regularization prevents the weights of the network from straying far from their intital values. When $\lambda \to \infty$, features are not updated and model operates in a lazy regime. We generate a sweep of pretrained models for $\lambda \in [0, 10^{-4}, 10^{-3}, 10^{-2}, 10^{-1}]$. We then linearly transfer each of these pretrained models to the target task and choose the model with the best generalization error (blue). The transferability degrades with target set size as expected, but the optimally regularized pretrained model avoids negative transfer, while the fully rich model (pink) transfers poorly for nearly all dataset sizes considered.

