# OpenReview forum: "Features are fate: a theory of transfer learning in high-dimensional regression"
_ICML.cc/2025/Conference — ICML 2025 poster_

### Official Review · Reviewer_jfTK · 2025-03-02

**Overall Recommendation:** 3

**Summary:**

The manuscript theoretically analyzes the transfer learning from a feature-centric viewpoint. Specifically, the authors consider the deep linear model and two transfer schemes, i.e. linear transfer and fine-tuning. Multiple theoretical results, such as phase diagrams, are established to uncover when transfer learning will outperform the training from scratch. Some insights are then extended to nonlinear networks slightly.

## update after rebuttal
The authors' additional results for finite-sample pre-training are interesting. Although I can't see the revised version, and the problem setting under the linear form is limited, I think this work will be a fair addition to the transfer learning community. Please make sure the finite-sample pre-training results will be added to the final version. As my original score is already positive, I will keep it as is.

**Claims And Evidence:**

The authors claimed those distributional measures between source and target data distributions are not enough to predict the success of transfer learning over training from scratch. This is also listed as one of the contributions of this manuscript. However, the authors showed that positive transfer can happen even if source and target distributions are far apart only in Appendix A. Since this is the motivation to consider the feature-centric viewpoint (and even the claimed contribution), the authors should not put it in the appendix.

**Essential References Not Discussed:**

I do not heavily work in transfer learning within over-parametrizing and feature learning regimes, but it seems like the authors have cited some reasonable reference in this field.

**Experimental Designs Or Analyses:**

Experimental designs seem correct, and the results seem to align with the theoretical results.

**Methods And Evaluation Criteria:**

The evaluation criteria are reasonable.

As for the training procedure of this paper, I’m not convinced. Specifically, the authors considered the pre-training over the source domain was conducted on the population distribution rather than empirical one, as the authors would like to mimic the practical setting where the sample size over the source domain greatly exceeds the sample size over the target domain. Therefore, the authors can show the pre-trained coefficient $\beta_{s}$ indeed converges to ground truth as $t\rightarrow\infty$.

However, I think this setting is problematic. It is unreasonable to assume one can train on the population distribution (or equivalent $n_{s} \rightarrow \infty$) and one can do the pre-training infinitely long. It is more reasonable to consider pre-training in which the source sample size is much larger than the target one ($n_{S}\gg n_{t}$), and the pre-trained model can not fully recover the source ground truth $t\neq \infty$. A lot of transfer learning theory works consider such cases where they can more precisely evaluate how sample sizes and model similarity can produce a positive transferability; see [1,2] and more references therein. Besides, since the pre-trained model can not perfectly recover the ground truth, there will be some extra error induced by adapting these features in linear transfer, which could change the phase diagrams greatly.

[1] Du, Simon S., et al. "Hypothesis transfer learning via transformation functions." *Advances in neural information processing systems* 30 (2017).

[2] Li, Sai, T. Tony Cai, and Hongzhe Li. "Transfer learning for high-dimensional linear regression: Prediction, estimation and minimax optimality." *Journal of the Royal Statistical Society Series B: Statistical Methodology* 84.1 (2022): 149-173.

**Other Comments Or Suggestions:**

The authors can probably be clearer about some of the notations. For example, explain what is $\bar{W}\_{l}$ and how to set it instead of just citing other papers, and state the signal strength $\||\beta\||_{2}$ equals 1.

**Other Strengths And Weaknesses:**

Strengths:

1. The manuscript is well-written and very easy to follow.
2. It is interesting to see how feature/model similarity can be used to predict the success of transfer learning, which is now gaining attention and investigation in the statistics community.
3. Results in Theorems 3.7, 3.8, and 3.9 seem novel in the field (transfer learning with deep linear network).

Weakness:

1. The pre-training process. Please refer to the “Methods And Evaluation Criteria" part.
2. The deep linear model is a very simple setting. Although I understand this may be due to the lack of available theoretical tools or techniques in the community to investigate the nonlinear network, this can limit how the theoretical analyses in this manuscript can provide insights into practical settings.
3. The argument for “distributional measure is not predictive of the success of transfer learning” is confined to only using KL divergence and Dudley Metric. It does not seem fully convincing, as there may be other measures/metrics.

**Questions For Authors:**

Questions:

1. Is it possible to make Theorem A.2 to be measure-free or for a broader class of distributional measure? I know this can be challenging, but it can make the motivation for a centric viewpoint convincing.
2. If the pre-training is not on population distribution, and the setting is $n_{s} \gg n_{t}$ and $t\neq \infty$, which part of your analyses will no longer hold?
3. Isn't the label shift referred to the case where the marginal distribution of $Y$ is unchanged across domains? I believe the case you studied is usually referred to as concept or model shifts.

**Relation To Broader Scientific Literature:**

The result of data similarity (distributional measure) might not be enough to predict the success of transfer learning and motivate the feature-centric viewpoint can produce a broader impact in transfer learning research.

**Theoretical Claims:**

I only went through the proofs roughly (which seem correct), but I did not check them in great detail.

---

> ### Author Rebuttal · Authors · 2025-04-01
>
> Dear Reviewer jfTK,
>
> Thank you for your thoughtful review of our work. We are grateful you found the manuscript easy to follow and we appreciate your feedback on areas that can be improved. We respond to your smaller comments below and leave our response to finite source datasets to the end, as this is more involved.
>
> First, thank you for pointing out these two references we have not cited. They will be included in the Related Works section of the introduction
> While we agree that deep linear networks are particularly simple, we chose to prioritize analytic tractability in this work. However, as we mention in our response to Reviewer 65EM, a lot of the take aways are the same in the case of linear transfer with nonlinear networks. In this case we can describe the feature space using Reproducing Kernel Hilbert Space theory as in Appendix D, where we carry out a calculation for two layer ReLu networks.
> You are correct that Theorem A.2 is proven only for the Dudley Metric and KL divergence. However, we note that there is a hierarchy of integral probability metrics. For example, the theorem also holds for The Wasserstein metric since it is lower bounded by the Dudley Metric. Although we expect it to be the case, we were not able to prove that the relation holds for any IPM or $\phi-$divergence.
> You are correct that we have called our model label shift, but it appears that the term “concept drift” is more common in the literature. We will adjust our terminology accordingly in the revised manuscript
> As for the question of finite source datasets, we agree that this is a richer setting to study transfer learning. To this end, we have carried out a calculation for the **full fine tuning** transferability surface in the case of a finite source dataset, the results of which are described in the following. Let $\gamma_s = n_s/d$ and $\gamma_t = \n_t/d$ where $n_s$ and $n_t$ are the number of source and target data points respectively. Let $\sigma_s$ be the standard deviation of the Gaussian label noise in the source task (defined analogously to Assumption 3.1). Then the transferability is
>
> $$
> \begin{cases}
> \frac{(\gamma_t - 1)}{1-\gamma_s}\sigma_s^2 \gamma_s + \gamma_s (\gamma_t -1) (1- 2\cos
> \theta) & \gamma_s, \gamma_t < 1,  \\
>  (\gamma_t -1)(1-2\cos\theta) -  \frac{(\gamma_t - 1)}{1-\gamma_s}\sigma_s^2 & \gamma_s > 1>\gamma_t , \\
> 0 & \gamma_t > 1
> \end{cases}
> $$
>
> The outline of the proof, which we will include in the final version of the paper, is similar in spirit to that of Theorem 3.9. In particular we know that gradient flow will converge to the minimum norm least square solution during source training, and we reason about the norm of its projection in the space orthogonal to the row space of the target data. There are a few interesting aspects of this expression. First, note that as long as the target task is overparameterized ($\gamma_t < 1$), the negative transfer boundary is completely determined by $\gamma_s$. Second, in the case of no label noise in the source task, the phase diagram is the same as in Fig 2. That is to say that fine tuning does not depend on the amount of source data if there is no label noise. The case of general noise level and source dataset size is more interesting: for some values of $\sigma_s$ there are disconnected regions of positive transfer. We will include the analysis of these equations, as well as plots of the phase diagram in the final version of the manuscript. We thank the reviewer for this suggestion.

---

> > ### Comment · Reviewer_jfTK · 2025-04-02
> >
> > Thank you for your response.
> >
> > I appreciate the part where you extend your results to pre-train over a finite source sample case. If there is room for the final revised version (if accepted), please ensure you add this **full fine-tuning** scenario in the main paper with sufficient discussion.
> >
> > Besides, citing the paper I mentioned is unnecessary as they are just examples of how I view the importance of finite sample pre-train. There is some work (to my knowledge) in this line that can be more related to your work. Authors can consider giving credit to such works.

---

### Official Review · Reviewer_AyJw · 2025-03-08

**Overall Recommendation:** 3

**Summary:**

This paper theoretically analyzes transfer learning under a multi-layer neural network model. The exact setting considered is a label shift setting with Gaussian noise and linear targets. The paper analyzes the features learned by the penultimate layer of the linear network and studies how the learned features relate to transferability. Simulations on two-layer ReLU networks trained in a mean-field regime show that the theoretical results on linear networks also transfer to non-linear ReLU networks.

**Post rebuttal update:** The authors' rebuttal addresses most of my concerns. While I still feel that the technical contribution of the paper is a bit limited, I agree that analyzing transfer learning with the inductive biases in feature learning considered, even in a simplified linear setting, holds some significance. I thus keep my score as is.

**Claims And Evidence:**

All claims made in the paper are supported by clear evidence.

**Essential References Not Discussed:**

To my knowledge, most of the related works are properly cited and discussed.

**Experimental Designs Or Analyses:**

Experimental designs are OK.

**Methods And Evaluation Criteria:**

The paper does not propose a new method. The experiments mainly serve as proof-of-concept demonstrations of theoretical results, which make sense to me.

**Other Comments Or Suggestions:**

- I think the first contribution listed in Lines 104-109 is a bit of an overclaim: as I have mentioned above, multi-layer linear networks are not a novel setting itself, so the claim "We develop an analytically solvable model of transfer learning that captures training dynamics, implicit bias, and generalization error in deep linear networks" feels too big for me.

- In the "Feature learning" paragraph of related work, I think several prior works should also be discussed (yet they do not belong to "essential references" so I did not list them above): [1] and [2] are among the first works theoretically analyzing the feature learning process beyond the neural tangent kernel regime. More related to transfer learning, [3] and [4] analyze the feature learning process of neural networks and its impact on generalization to new test distributions.

- Lines 271-273: "This condition requires that there is more data than the there is target function power in the direction learned during pretraining." (non-comprehensible sentence)

---

[1] Allen-Zhu et al. Towards Understanding Ensemble, Knowledge Distillation and Self-Distillation in Deep Learning. arXiv, 2020.

[2] Allen-Zhu et al. Feature Purification: How Adversarial Training Performs Robust Deep Learning. FOCS, 2021.

[3] Chen et al. Understanding and improving feature learning for out-of-distribution generalization. NeurIPS, 2023.

[4] Zhang et al. Feature contamination: Neural networks learn uncorrelated features and fail to generalize. ICML, 2024.

**Other Strengths And Weaknesses:**

**Strengths:**
- The theoretical parts are well-written.
- Simulations are comprehensive and some results are interesting to me.

**Weaknesses:**
- The technical novelty is limited. Multi-layer linear networks are a rather well-understood setting with much prior work, and the paper mainly uses the results in prior work to further analyze transfer learning.

**Questions For Authors:**

Based on the transferability analysis, can you comment on the benefits of linear probing/fine-tuning? For example, when should we choose fine-tuning/linear probing for transfer learning?

**Relation To Broader Scientific Literature:**

Most of the theoretical results of this work are built upon Yun et al. (2021): since the trained model can be fully characterized in the considered multi-layer linear network setting, analyzing the generalization error/transferability seems rather straightforward based on the trained model. That being said, I think the transferability results presented are a decent addition to the existing literature. I also found the results showing the insufficiency of distributional source-target measures interesting.

**Theoretical Claims:**

I did not carefully check the proofs. Yet, many results (e.g., Theorem 3.4 and 3.5) are built on prior work (Yun et al., 2021), and based on my understanding of the problem, they seem correct.

---

> ### Author Rebuttal · Authors · 2025-04-01
>
> Dear Reviewer AyJw,
>
> Thank you for your review of our work. We are glad you found the paper well written and we thank you for your feedback on how to improve the manuscript. First, we appreciate your pointing out the additional references. We will discuss their relevance to our work in the Related Works section of our updated draft. Thank you for pointing out moments that felt unclear or overstated. In regards to Lines 104-109 we primarily want to highlight that training dynamics, implicit bias, and generalization error are all relevant to a theory of transfer learning and that deep linear models are an ideal test bed to explore the interplay of these phenomena since their manifestations are analytically solvable. We agree that many works have explored deep linear networks, and we will change these lines to highlight that our results build on prior work to tackle these multifaceted aspects of the transfer learning problem. Thank you for also pointing out the lack of clarity in Lines 271-273. We will change this sentence to: “We can view $\gamma$ as a dimensionless measure of the amount of target data, and $\cos^2 \theta$ as the amount of power that the target function has in the subspace of the pretraining task. The condition for negative transfer is satisfied when there is more target data than there is power in the pretrained subspace”. In response to your final question, there indeed is a regime where linear probing will outperform fine tuning. We can solve for this condition by subtracting Equation 15 from Equation 11 and looking for points in the $(\theta, \gamma)$ plane where this function is positive. In the noiseless case, the expression simplifies a bit. In the overparameterized regime ($\gamma < 1$) linear probing out performs fine tuning as long as $\gamma < \sin^2(\theta/2)$. The intersection of this condition with that for positive transfer ($\gamma < \cos^2{\theta}$) is satisfied when there is limited target data. We will include a plot of this region in the appendix of the final draft. For the underparameterized case ($\gamma > 1$) linear probing always induces negative transfer whereas fine tuning has zero transferability. So, although fine tuning has better transferability than linear probing, it’s not worth doing any pretraining in the underparameterized regime if there is no label noise.

---

> > ### Comment · Reviewer_AyJw · 2025-04-03
> >
> > Thank you for your response, which addresses most of my concerns. While I still feel that the technical contribution of the paper is a bit limited, I agree that analyzing transfer learning with the inductive biases in feature learning considered, even in a simplified linear setting, holds some significance. I thus keep my score as is.
> >
> > - On the comparison between linear probing and fine-tuning: Thank you for the response on this point. I think the dependency of the advantage of linear probing/fine-tuning on the learning setting is indeed interesting. Perhaps the authors could also consider discussing the relation between your results and those in [1], which also compares linear probing with fine-tuning in an overparameterized setting and has been cited in your paper, but seems without detailed discussion.
> >
> > ---
> > [1] Kumar et al. Fine-tuning can distort pretrained features and underperform out-of-distribution. ICLR, 2022.

---

### Official Review · Reviewer_UEjP · 2025-03-13

**Overall Recommendation:** 4

**Summary:**

The paper analyzes the transferability capability of deep linear networks. Specifically, it theoretically analyzes the generalization error of deep linear networks when they are trained from scratch versus linear transfer and fine-tuning in a regression problem. The paper also extends this study to the use of the ReLU activation function. In particular, it focuses on a theoretical analysis of performance based on a feature space learned by the pre-trained model. The aim is to evaluate when a model can be beneficial, for which the theoretical assumptions consider a mathematically tractable model to study transfer learning, introducing deep linear networks, which, although a simplification, capture how neural networks learn features.
The findings of this article demonstrate that transfer improves performance if the overlap of the feature space between the source task and the target task is sufficiently strong. Furthermore, approaches such as linear transfer and fine-tuning are compared, showing that the efficiency of transfer depends on the structure of the features learned during pretraining.

## update after rebuttal
After reading other reviews and responses, I prefer to keep my score. There are no critical points to decrease/increase my evaluation.

**Claims And Evidence:**

Yes, they are.

**Essential References Not Discussed:**

No, I can not think of any paper that must be included.

**Experimental Designs Or Analyses:**

Yes, the analysis seems fine.

**Methods And Evaluation Criteria:**

Yes, they make sense. The paper is theoretical, and some empirical examples are used to corroborate the theory.

**Other Comments Or Suggestions:**

-While the article is based on a robust theoretical analysis, it would be helpful to include a more detailed section on experiments or simulations that support the theoretical conclusions, providing more concrete validation of the approach.

**Other Strengths And Weaknesses:**

Strengths:
-The writing is clear, and the theoretical formulations are well explained, allowing the article to be understood even by readers without deep knowledge of the topic. The logical organization of the sections also facilitates comprehension, and the article is presented in an accessible manner, maintaining an appropriate balance between formality and clarity.
-The choice of a linear network-based approach simplifies the mathematical formulation and facilitates the demonstration of phenomena related to model transfer. This theoretical approach provides clarity in the concepts and makes the article accessible to a broader audience.
-The article provides a detailed analysis of the conditions for successful transfer, especially by considering the similarity between the source and target tasks. It also clearly discusses whether the representations learned in both domains are relevant, which is essential for transfer success. This analysis is a strong point, as it helps to connect ideas with practical situations in Transfer Learning.

Weaknesses:
-Equation (6), a parenthesis is missing in f(x)^2?

**Questions For Authors:**

No comments or questions

**Relation To Broader Scientific Literature:**

It's quite important. Linear transfer and fine tuning are some of the most used methods in the literature. Unfortunately, people use them without any knowledge and do not understand that sometimes, the model can do even worse. Even though the paper makes several assumptions, this can be a very important theoretical contribution.

**Theoretical Claims:**

No, I did not have time to analyze the proofs corresponding to Appendix C.

---

> ### Author Rebuttal · Authors · 2025-04-01
>
> Dear Reviewer UEjP,
>
> Thank you for taking the time to review our paper. We are happy to hear that you found our results interesting and the presentation clear. We agree that we could be more detailed in our presentation of the simulations. In the final draft of the manuscript we plan to make the code publicly available, including a jupyter notebook that readers can use to explore the concepts of the paper for themselves. In addition, we will mention Appendix E in the main text, per your recommendation.

---

### Official Review · Reviewer_65EM · 2025-03-14

**Overall Recommendation:** 4

**Summary:**

The authors develop a a feature-centric theory of transfer learning, based on their insight that transferability is a property of the learned
feature space and not only of the source and target datasets.
The theory is developed for a deep linear networks and analytically characterizes the transferability phase diagram as a
function of the target dataset size and the feature space overlap.
The authors report a few insights based on their theoretical analysis, and demonstrate potential applicability in two-layer ReLU networks

**Claims And Evidence:**

Paritally.  I am unsure about the claim about lazy vs rich training mentioned in Section 5, and in generally,  claims related to training dynamics.

**Essential References Not Discussed:**

Please cite the following reference for lazy vs rich training:

Andrew M. Saxe, James L. McClelland, and Surya Ganguli. Exact solutions to the nonlinear dynamics of learning in deep linear neural networks. ICLR 2014

**Experimental Designs Or Analyses:**

N/A. The analysis is mainly theoretical.

**Methods And Evaluation Criteria:**

Yes

**Other Comments Or Suggestions:**

A few typos:
- intital => initial
- discrepency => discrepancy
- and model operates => the model
- Gram matrix => always capitalize

**Other Strengths And Weaknesses:**

The authors provide many profound insights, however:
- Taken strictly, the results are limited to deep linear networks.
- The parctical implications are not clear.
- The manuscript is rather difficult to follow. I understand that this is a theory paper, but I would appreciate including an example that guides the reader through the theorems and what they mean for that example.
- A visualization of the feature space would be very helpful, especially to explain the overlap.
- I miss  the number of epochs and learning rate as influential parameters that impacts transferability in the rich regime.

**Questions For Authors:**

- How do you measure feature space overlap ($\theta$)? And why you use radian as its unit?

- I found it surprising to see a goldilock effect w.r.t. target dataset size ($\lambda$), as illustrated in Figure 1. Is this because a large target set negates any benefits of transfer learning and tip the scale towards training from scratch?

- Do the results extend beyond two-layers in ReLU networks?

**Relation To Broader Scientific Literature:**

Sufficient novelty

**Theoretical Claims:**

Partially

---

> ### Author Rebuttal · Authors · 2025-04-01
>
> Dear Reviewer 65EM,
>
> Thank you for your thoughtful review and feedback. You are correct that the rigorous proofs are limited to deep linear networks. This architecture exhibits certain symmetries which generate a conserved quantity in gradient flow that we exploit heavily to prove convergence. However, we believe the main takeaway of the work is applicable to other architectures, as demonstrated by the experiments in shallow nonlinear networks. Namely, during gradient descent in the feature learning regime, the model will learn features present in the source task. This defines a subspace of functions that the model can represent. When transferring to a target task with linear transfer, the model can only fit the portion of the target function that lives in the space spanned by the learned features of the target task. Since we can show this effect precisely in the deep linear setting we focus on this as the primary running example in the manuscript. The main practical take-away is that feature space overlap is the relevant object for predicting transferability. The focus of this paper is to demonstrate this phenomenon precisely in a solvable model, but we believe that interesting directions for future work include designing feature-level metrics that are predictive of transfer performance, or designing optimal sampling protocols for the source task when source sampling is readily available, but target data is scarce. Per your request of practical implications, we will add these ideas to our discussion section. While the space constraints keep us from including a simple example in the main body of the paper, we plan to make our code publicly available, including a Jupyter notebook with a simple example to help readers explore the concepts of our work. Additionally, we will include an additional figure with a cartoon depiction of the feature space overlap concept in the main body of the paper. We also thank you for raising the question of finite training time and learning rate. We show in Appendix C2 that in deep linear networks that convergence to a global minimum is subexponential in time, so at long times the results of our paper hold with very small error. However, optimal early stopping is an interesting and likely useful regularization technique that would prevent the model from sparsifying to the source features. We predict that the results of source task early stopping would be similar to those for weight decay, which we demonstrate in the appendix (Figure 5). That is, keeping the model from completely learning the source task may actually help transfer if the source and target features are very different. As for the learning rate, we believe that this framework would also describe models trained with other optimization algorithms such as finite size gradient descent or SGD. The challenge lies in describing their dynamics precisely. For this reason, we focused on the analytically tractable setting of gradient flow. However, similar theorems on the global convergence of gradient descent for finite step size are proven in “Global Convergence of Gradient Descent for Deep Linear Residual Networks” (Wu 2019). The main takeaway is that with a sufficiently small learning rate the network converges to a global optimum, in which case our results would hold exactly as written in the paper. Finally, we respond to your specific questions below:
>
> We define the feature space overlap in the following way: during the source training the model will learn a function $f = \sum_{i} c_i \phi_i{x}$. We call span($\{\phi_i\}$) the feature space of the source task. The feature space overlap is the norm of the orthogonal projection (in the $L_2$ sense) of the target function $f_t$ into this space. In the case of our model, the learned feature space is $\beta_{s}$ (see Theorem 3.4), the projection is simply $\beta_{s}^T \beta{t}$. Since both vectors have unit norm, this can be viewed as the cosine angle between the two tasks, so we plot the overlap in radians. In the case of two layer ReLU networks, computing the overlap is also analytically tractable. We describe how this is done in Appendix D.
> That is the correct intuition. We measure the success of transfer learning by its generalization relative to training from scratch on the target dataset. If the target dataset size is very large, there is little benefit to pretraining.
> Yes we believe our results extend beyond two layer ReLU networks. Equation 18 holds generally for any Reproducing Kernel Hilbert Space. We choose two layer ReLU networks with Gaussian data since the projection in Equation 18 can be computed exactly.

---

> > ### Comment · Reviewer_65EM · 2025-04-05
> >
> > I appreciate the detailed response by the authors which clarified many issues.
> > I will raise my score to 4 accordingly.

---

### Decision · Program_Chairs · 2025-05-01

**Decision:**

Accept (poster)

**Comment:**

The submission focuses on the relevance of feature representation as key element to determine the success of transfer learning. It has been overall appreciated for its novelty in content and its contribution in shedding some light on the role of features in such a learning strategy. The analysis provides in fact a theoretical framework to a number of practices (such as fine tuning), although it is focused a minimal theoretical model (a possible weakness of the submission) that might be far from real-world setup. Nevertheless, the shared positive impression leads to me to suggest to *Accept* the manuscript in ICML.